# A mast cell-ILC2-Th9 pathway promotes lung inflammation in cystic fibrosis

Silvia Moretti[1], Giorgia Renga[1], Vasilis Oikonomou[1], Claudia Galosi[1], Marilena Pariano[1], Rossana G. Iannitti[1], Monica Borghi[1], Matteo Puccetti[1], Marco De Zuani[2], Carlo E. Pucillo[2], Giuseppe Paolicelli[1], Teresa Zelante[1], Jean-Christophe Renauld[3], Oxana Bereshchenko[4], Paolo Sportoletti[5], Vincenzina Lucidi[6], Maria Chiara Russo[7], Carla Colombo[7], Ersilia Fiscarelli[8], Cornelia Lass-Flörl[9], Fabio Majo[6], Gabriella Ricciotti[8], Helmut Ellemunter[10], Luigi Ratclif[11], Vincenzo Nicola Talesa[1], Valerio Napolioni[1] & Luigina Romani[1]

T helper 9 (Th9) cells contribute to lung inflammation and allergy as sources of interleukin-9 (IL-9). However, the mechanisms by which IL-9/Th9 mediate immunopathology in the lung are unknown. Here we report an IL-9-driven positive feedback loop that reinforces allergic inflammation. We show that IL-9 increases IL-2 production by mast cells, which leads to expansion of CD25$^+$ type 2 innate lymphoid cells (ILC2) and subsequent activation of Th9 cells. Blocking IL-9 or inhibiting CD117 (c-Kit) signalling counteracts the pathogenic effect of the described IL-9-mast cell-IL-2 signalling axis. Overproduction of IL-9 is observed in expectorates from cystic fibrosis (CF) patients, and a sex-specific variant of IL-9 is predictive of allergic reactions in female patients. Our results suggest that blocking IL-9 may be a therapeutic strategy to ameliorate inflammation associated with microbial colonization in the lung, and offers a plausible explanation for gender differences in clinical outcomes of patients with CF.

[1] Department of Experimental Medicine, University of Perugia, 06132 Perugia, Italy. [2] Department of Medical and Biological Science, University of Udine, 33100 Udine, Italy. [3] Ludwig Institute for Cancer Research, Brussels Branch, B-1200 Brussels, Belgium. [4] Department of Medicine, Section of Pharmacology, University of Perugia, 06132 Perugia, Italy. [5] Institute of Haematology-CREO (Centro di Ricerche Emato-Oncologiche), Ospedale S. Maria Misericordia, 06132 Perugia, Italy. [6] Unit of Endocrinology and Diabetes, Bambino Gesù Children's Hospital, 00165 Rome, Italy. [7] Fondazione IRCCS Ca' Granda, Ospedale Maggiore Policlinico, University of Milan, 20122 Milan, Italy. [8] Bambino Gesù Children's Hospital IRCCS, 00165 Rome, Italy. [9] Division of Hygiene and Medical Microbiology, Innsbruck Medical University, 6020 Innsbruck, Austria. [10] CF Centre, Medical University Innsbruck, 6020 Innsbruck, Austria. [11] Servizio di Supporto Fibrosi Cistica, Istituto Ospedale G. Tatarella, Foggia, 71042 Cerignola, Italy. Correspondence and requests for materials should be addressed to L.R. (email: luigina.romani@unipg.it).

Innate lymphoid cells (ILCs) perform a variety of immune functions at barrier surfaces[1]. Three types of ILCs have been reported, which differ on the basis of the cytokines produced. ILC1 encompass natural killer cells and interferons (IFN)-γ-releasing cells; ILC2 release IL-5, IL-9 and IL-13, and ILC3 release IL-17A and IL-22. ILC2 preferentially localize to the interface between the host and the environment (lung, intestine and skin) and perform a variety of biological functions in mice[2] and humans[3]. In the lung, ILC2 and their cytokines play pro-inflammatory roles in allergic inflammation[2,4,5], but also protective roles in airway epithelial cell repair and control of tissue inflammation linked to pathogens[6,7]. Thus, ILC2 may affect the course of airways diseases, resulting in either pathological or protective outcomes. Lung ILC2 rapidly produces IL-5 and IL-13 on exposure to IL-33 (ref. 5), an effect potentiated by IL-25 and thymic stromal lymphopoietin (TSLP)[5], and IL-9 on the exposure to IL-2 (ref. 8). By promoting ILC2 survival[8], IL-9 provides a positive feedback loop that amplifies ILC2 cytokine production and the ensuing allergic airway inflammation[9]. However, IL-9 also dampens the pathogenic activities of Th17 cells[10] and mediates tolerance imparted by regulatory T cells (Treg) via mast cells (MC)[11]. Produced by MC, in addition to ILC2 and Th9, IL-9 in turn affects the expansion[12] and function[13] of MC, which are known to have positive, as well as negative, immunomodulatory roles in vivo[13–16]. Thus, IL-9, like ILC2, may have different roles in lung immune homeostasis.

In patients with cystic fibrosis (CF), the primary source of morbidity and mortality is due to a vicious cycle of airway infection and inflammation eventually resulting in lung damage. The inflammatory response in CF is dysregulated at several levels, resulting in inefficient microbial clearance and contributing to lung damage[17]. This is supported by several studies that have documented an altered balance of inflammatory/anti-inflammatory cytokines in CF (ref. 17), providing evidence that targeting specific inflammatory/anti-inflammatory pathways is a valid therapeutic strategy in CF (ref. 18). This balance is essential for the efficient control of Aspergillus fumigatus diseases in CF (ref. 18), where the colonization by the fungus is common and may lead to fungal sensitization, bronchitis and allergic broncho-pulmonary aspergillosis (ABPA)[19] as well as worse forced expiratory volume in the first second (FEV1) (ref. 20). In CF patients, the expression of IL-9 and IL-9R is increased and is associated with mucus overproduction, but whether and how IL-9 contributes to immunity and pathology in response to the fungal infection in CF is not known.

In the present study, we determine the contribution of IL-9 to Aspergillus infection and allergy in murine and human CF, and assess the therapeutic effectiveness of targeting IL-9-dependent pathways and the diagnostic potential of this approach. We find that IL-9-driven IL-2 production by MC expands $CD25^+$ ILC2, which in turn activate Th9 cells, leading to an amplified allergic inflammation. Overproduction of IL-9 is observed in expectorates from CF patients and a genetic variant of IL-9 shows a sex-specific association with IgE levels in female patients. Blocking IL-9 or inhibiting CD117 (c-Kit) signalling counteracts the pathogenic potential of the IL-9-MC-IL-2 axis, thus providing a therapeutic angle to ameliorate the pathological consequences of microbial colonization in CF.

## Results

### IL-9 production and ILC2-Th9 activation during aspergillosis.
We infected C57BL/6 or $Cftr^{-/-}$ mice intranasally with A. fumigatus and measured IL-9 production, ILC2 and Th9 cell activation in infection. We have already shown that $Cftr^{-/-}$ mice are susceptible to Aspergillus infection (from $2.5 \pm 0.7$ to $3.9 \pm 1.0$ log colony forming unit (cfu) ± s.d. per lung, C57BL/6 versus $Cftr^{-/-}$ mice, respectively) and allergy (from $9.2 \pm 0.7$ to $22.4 \pm 1.3$ ng ml$^{-1}$ total serum IgE in C57BL/6 and $Cftr^{-/-}$ mice, respectively). A peak production of IL-9 occurred during the first week of the infection in C57BL/6 mice to decline thereafter as opposed to $Cftr^{-/-}$ mice in which levels of IL-9 were sustained throughout the infection (Fig. 1a). Peak production of IL-9 was also observed in $Rag1^{-/-}$, and less in $Rag1^{-/-}/Il9R^{-/-}$, mice early but not late in infection (Fig. 1a), a finding suggesting that early IL-9 production is IL-9R-dependent and late is T-cell-dependent. We looked therefore for the presence of IL-9$^+$ILC2 and Th9 cells in infection by characterizing IL-9-producing Lin$^-$ and CD4$^+$ T cells in the lung. ILC2 are marked by expression of the IL-33R as well as the common γ chain (γc) cytokine receptors for IL-2 and IL-7 (ref. 2). Flow cytometry analysis revealed that CD90.2$^+$ILC2 expressing IL33R or CD25 were present in the lung of naive C57BL/6 (4.5 and 3.3% for CD25$^+$ and ST2$^+$ cells, respectively) and $Cftr^{-/-}$ mice (5.8 and 4.0% for CD25$^+$ and ST2$^+$ cells, respectively; Fig. 1b,c). In C57BL/6 mice, and similarly in $Rag1^{-/-}$ mice (Supplementary Fig. 1a), ST2$^+$ILC2 cells decreased early in infection to return to baseline level 10 days later while CD25$^+$ILC2 stably decreased (Fig. 1b,c). In contrast, in $Cftr^{-/-}$ mice, both types of ILC2 steadily increased throughout the infection (Fig. 1b) along with the expression of the ILC2 transcription factors, Rora, and Gata3 (Fig. 1d) and the production of ILC2 effector cytokines, IL-5 and IL-13 (Fig. 1e). IL-9-producing CD90.2$^+$ILC2 were also expanded in $Cftr^{-/-}$ mice but not in C57BL/6 (Fig. 1b,c) and $Rag1^{-/-}$ mice (Supplementary Fig. 1a), as revealed by flow cytometry. In terms of Th9 cell activation, CD4$^+$IL-9$^+$ T cells appeared in C57BL/6 mice a week after the infection to decline thereafter (Fig. 1h), consistent with the short retention of Th9 at the inflammatory sites[21]. The expansion was instead sustained in $Cftr^{-/-}$ mice (Fig. 1h) along with the expression of Il9, Pu.1 (purine-rich box 1) and Irf4 (interferon regulatory factor 4) transcription factors (Fig. 1g). These data indicate that IL-9$^+$ILC2 and Th9 cells are all increased in $Cftr^{-/-}$ mice during A. fumigatus infection.

Given that ILC1 through IFN-γ (ref. 22) and ILC3 through IL-22 (ref. 23) may affect ILC2 expansion, the differential expansion of ILC2 could reflect the ILCs dynamics in the lung. However, NKp46$^+$NK1.1$^+$ ILC1 cells producing IFN-γ did not expand and ILC1-promoting cytokines IL-15 and IL-18 were not produced in $Cftr^{-/-}$ as opposed to C57BL/6 mice (Supplementary Fig. 2a,b). Similarly, despite expanded in $Cftr^{-/-}$ mice, CCR6$^+$RORγt$^+$ ILC3 produced IL-17A more than IL-22 (Supplementary Fig. 2c,d). Thus, while confirming the defective production of IFN-γ and IL-22 in $Cftr^{-/-}$ mice[18], these findings suggest that the expansion of ILC2 in $Cftr^{-/-}$ mice is not dependent on ILCs dynamics in the lung but rather on the production of ILC2 promoting cytokines. This appeared to be the case, as the levels of cytokines promoting ST2$^+$ILC2, IL-33 and CD25$^+$ILC2, IL-2, were constantly elevated in $Cftr^{-/-}$ mice whereas a peak production was only observed at an early time point in C57BL/6 mice (Fig. 1f).

### IL-9 contributes to inflammatory pathology in infection.
To assess the role of IL-9 in A. fumigatus infection and allergy, we resorted to $Il9R^{-/-}$ mice that, given the crucial role of the IL-9R, a member of the γc receptor family, for the survival of lung ILC2 (ref. 8), also have a decreased ILC2 (ref. 8). Mice were either acutely infected with Aspergillus conidia intranasally or subjected to fungal allergy (ABPA) by repeated sensitization with Aspergillus culture filtrate extracts ($5.5 \pm 0.7$ ng ml$^{-1}$ versus $11 \pm 1.0$, total serum IgE in $Il9R^{-/-}$ versus C57BL/6 mice,

respectively). We found that the absence of IL-9R signalling conferred resistance to both infection and allergy, as indicated by the reduced fungal load (Fig. 2a) and decreased inflammatory lung pathology in infection as well as in ABPA (Fig. 2b). The numbers of lung CD25[+] and ST2[+] ILC2 were decreased in these mice as revealed by immunofluorescence staining (Fig. 2c). Concomitantly, Th9 and, partially, Th2 cells—revealed by *Stat6* expression and STAT5 phosphorylation (Supplementary Fig. 1b,c)—were decreased in both infection and allergy (Fig. 2d,e), while Th17 and Treg cells were unaffected and Th1 cells increased (Supplementary Fig. 3a,b). Corroborating these findings, IL-9 neutralization in C57BL/6 or *Cftr*[−/−] mice greatly ameliorated lung pathology in response to the fungus, both in

terms of inflammatory cell recruitment (Fig. 2f) and fibrosis as shown by Masson's trichrome staining (insets of Fig. 2f) and production of TGF-β (Fig. 2g), a mediator of pulmonary fibrosis[24]. Together, these results indicate that the IL-9/IL-9R signalling pathway is required for the expansion of pathogenic ILC2 and Th9 cells in response to the fungus. However, whether ILC2 promote Th9 cell activation via IL-9R signalling is not known. To directly assess this, we did criss-cross experiments in which CD4[+] T cells from either C57BL/6 or *Cftr*[−/−] mice were assessed for IL-9 production and Th9 transcription factor expression on co-cultivation in a transwell permeable support with lung Lin[−] cells exposed to *A. fumigatus* and either IL-2 or IL-33. We found that Th9 cell activation was observed upon

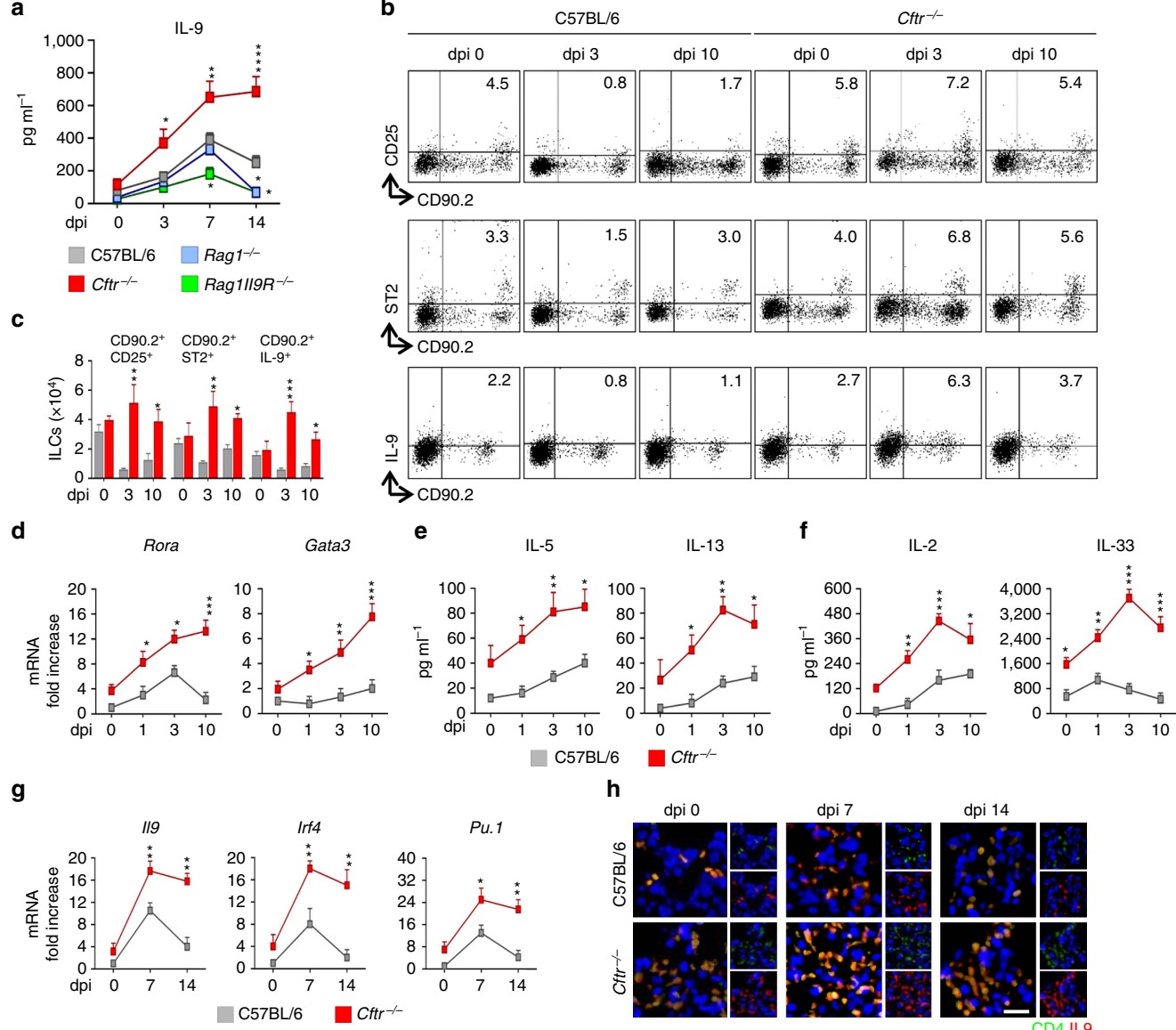

**Figure 1 | IL-9 production and ILC2-Th9 cells activation in *Aspergillus fumigatus* infection.** (**a**) Time course of IL-9 production at various days post infection (dpi) in mice (six per group) infected intranasally with live *A. fumigatus* conidia. (**b**) Detection of CD90.2[+]CD25[+], CD90.2[+]ST2[+] and CD90.2[+]IL-9[+] lung type 2 ILCs by flow cytometry (numbers refer to percentages of positive cells) and immunofluorescence staining. (**c**) Absolute number of lung ILC2; (**d**) ILC2–specific transcript on lineage negative lung cells; (**e,f**) ILC2 effector and activating cytokines; (**g**) *Il9* and Th9-cell specific transcripts on lung CD4[+] T cells and (**h**) immunofluorescence staining of lung CD4[+] IL-9[+] T cells. Photographs were taken with a high-resolution microscope (Olympus DP71) equipped with a ×40 objective; scale bar, 100 μm. Mean values ± s.d. cytokines were determined on lung homogenates by ELISA, *Il9* and transcripts assessed by PCR with reverse transcription. 0, uninfected mice. *P < 0.05, **P < 0.01, ***P < 0.001, ****P < 0.0001, knockout versus C57BL/6 mice (data represent pooled results or representative images from three experiments, Two-way ANOVA, Bonferroni post test). *Gata3*, GATA binding protein 3; *Irf4*, interferon regulatory factor 4; *Pu.1*, purine-rich box 1; *Rora*, RAR-related orphan receptor alpha.

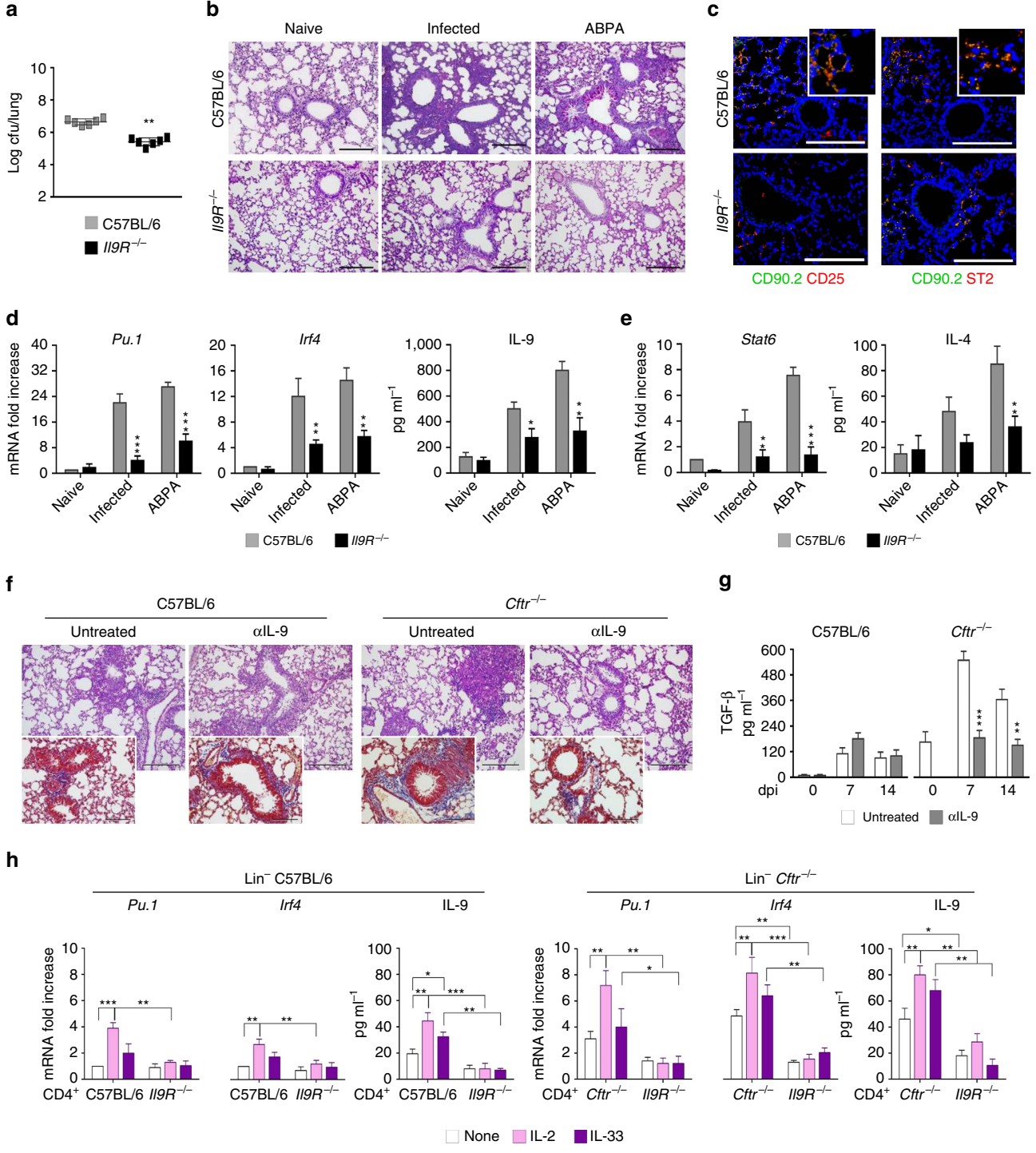

**Figure 2 | IL-9R signaling contributes to inflammation and allergy.** C57BL/6 and $Il9R^{-/-}$ mice (six per group) were intranasally infected with live *Aspergillus fumigatus* conidia or subjected to ABPA and assessed for (**a**) lung fungal growth (log₁₀ cfu, mean ± s.d.); (**b**) lung histology (periodic acid − Schiff staining); (**c**) expression of CD90.2⁺CD25⁺, CD90.2⁺ST2⁺ lung ILC2 by immunofluorescence; (**d,e**) Th-cell specific transcripts and cytokine production. (**f**) Lung histology (periodic acid–Schiff and, in the inset, Masson's trichrome staining) and (**g**) TGF-β production in C57BL/6 or $Cftr^{-/-}$ mice infected as above and treated with IL-9 neutralizing antibody for a week. Days post infection (dpi). (**h**) Th-cell specific transcripts and IL-9 production of lung CD4⁺ T cells from naive mice co-cultured with lung lineage negative (Lin⁻) cells in the presence of *A. fumigatus* conidia, IL-2 or IL-33. Photographs were taken with a high-resolution microscope (Olympus DP71) equipped with a ×20 objective; scale bars, 200 μm and a ×40 objective (insets of **f**, scale bars, 100 μm). Results are mean values ± s.d., ELISA was done on lung homogenates and culture supernatants for cytokines and PCR with reverse transcription on CD4⁺ lung cells. *$P < 0.05$, **$P < 0.01$, ***$P < 0.001$, $Il9R^{-/-}$, $Cftr^{-/-}$ versus C57BL/6 mice; IL-9-treated versus control isotype-treated mice; stimulated versus unstimulated (none) cells and $Il9R^{-/-}$ versus C57BL/6 or $Cftr^{-/-}$ CD4⁺ T cells. Naive, uninfected mice. Data represent pooled results or representative images from three experiments, Two-tailed Student's t-test (**a**) or Two-way ANOVA (**d,e**) Bonferroni post test. *Gata3*, GATA binding protein 3; *Irf4*, interferon regulatory factor 4; *Pu.1*, purine-rich box 1.

co-cultivation of CD4$^+$ T cells with Lin$^-$ cells in the presence of IL-2 more than IL-33, an effect magnified in $Cftr^{-/-}$ as compared to C57BL/6 mice (Fig. 2h), and requiring the presence of IL-9R on responder CD4$^+$ T cells being significantly negated with $Il9R^{-/-}$ responder cells (Fig. 2h and Supplementary Fig. 4). These results indicate that ILC2, and particularly CD25$^+$ILC2, may account for the sustained IL-9 production and Th9 activation responsible for pathology in CF.

**IL-9 activates mast cells to produce IL-2**. The sustained production of IL-33 and IL-2 in $Cftr^{-/-}$ mice prompted us to investigate mechanisms behind this production. IL-33 is constitutively expressed at epithelial barrier surfaces where it is rapidly released from cells during tissue injury[25]. However, tight regulation of IL-33 following its release to dampen ST2-dependent inflammation to fungi has also been described[26]. This likely occurred in C57BL/6 but not in $Cftr^{-/-}$ mice in which the high levels of epithelial damage observed upon the infection (Fig. 3a) likely accounted for the sustained IL-33-dependent ST2$^+$ILC2 expansion. For IL-2, predictably high in CF, given the sustained NFAT activity[27], in addition to CD4$^+$ T cells[8] and dendritic cells[28], MC are known to produce it in the lung[7]. We looked therefore for MC presence and activity in the lung of C57BL/6 and $Cftr^{-/-}$ mice after the infection. MC were much expanded in $Cftr^{-/-}$ mice, as seen by flow cytometry (Fig. 3b) and toluidine staining (Fig. 3c). MC are distinguished by their granule content whose expression is tissue-dependent[29]. In the lung, chymase-positive MC numbers positively correlated with better lung function[30], whereas chymase- and tryptase-positive MC were expanded in areas of fibrosis in CF lungs and positively correlated with the degree of fibrosis and lung function[31]. Immunohistochemistry revealed that while tryptase-positive cells could not be detected, chymase-positive MC were present in C57BL/6 mice (insets of Fig. 3c). In contrast, chymase-positive and tryptase-positive MC were observed in $Cftr^{-/-}$ mice (insets of Fig. 3c).

As MC are known to produce cytokines through different receptor mechanisms[32], we evaluated cytokine production on magnetically purified c-Kit$^+$ MC (as characterized by morphometry and MC specific transcripts, Fig. 3d) upon stimulation with IgE, IL-9 or IL-33. We found that IL-2 production was induced by IL-33 and, more, by IL-9 and not by IgE, mostly in MC from $Cftr^{-/-}$ mice (Fig. 3e). As IL-9 also induced IL-9 production (Fig. 3e) and IL-9$^+$MC could also be detected *in vivo* (Fig. 3b), this suggests that an autocrine loop appears to mediate the IL-9-dependent IL-2 release by MC. As a matter of fact, IL-2 production ($39 \pm 6$ ng/ml versus $127 \pm 22$, IL-2 in lung homogenates at 3 dpi in $Il9R^{-/-}$ versus C57BL/6 mice, respectively) and IL-2$^+$MC (Fig. 3f) were greatly reduced in $Il9R^{-/-}$ mice, thus contributing to the defective expansion of CD25$^+$ILC2 in these mice. Of interest, IL-9 stimulation also induced TGF-β in MC from $Cftr^{-/-}$ mice but not IL-6 (Fig. 3e), a finding suggesting that the autocrine IL-9 stimulation appears to be specific for IL-2 and TGF-β (Fig. 3e). These data indicate that MC may contribute to IL-2 production eventually leading to CD25$^+$ILC2 expansion in $Cftr^{-/-}$ mice. This appears to be the case, as IL-2$^+$MC, more than IL-2$^+$CD90.2 or IL-2$^+$CD4$^+$ T cells, were expanded *in vivo*, early in infection, particularly in $Cftr^{-/-}$ mice (Fig. 3f).

To directly prove this, we assessed susceptibility to inflammatory allergy of MC-deficient C57BL/6-$Kit^{W/W-v}$ mice or $Cftr^{-/-}$ mice treated with the tyrosine kinase inhibitor imatinib known to inhibit IL-9-driven mastocytosis in the lung[12]. Airway mastocytosis was reduced in MC-deficient $Kit^{W/W-v}$ mice (Fig. 4a) along with reduced levels of IgE, IL-2, IL-9 and

TGF-β (Fig. 4b). Concomitantly, the number of CD25$^+$ILC2 were also decreased in the lung but promptly restored upon MC engraftment or exogenous IL-2 administration (Fig. 4a). Thus, MC appear to be able to control CD25$^+$ILC2 expansion in the lung during the infection via IL-2. Similar results were obtained upon treatment of $Cftr^{-/-}$ mice with imatinib. Both inflammation (Fig. 4c), collagen deposition (insets of Fig. 4c), IL-2, IL-9 and TGF-β production (Fig. 4d) and Th9 cell activation (Fig. 4e) were attenuated. Interestingly, imatinib apparently increased early inflammation in C57BL/6 mice (Supplementary Fig. 5), a finding suggesting that c-Kit$^+$ cells could contribute to pathogen resistance early in infection. As a matter of fact, MC-deficient $Kit^{W/W-v}$ mice displayed increased susceptibility to the infection as compared with C57BL/6 mice, as indicated by the increased fungal load (Supplementary Fig. 6a), neutrophils recruitment (Supplementary Fig. 6b) and a degree of lung inflammation (Supplementary Fig. 6c).

**CFTR deficiency contributes to inflammation**. The above results suggest that a circuit involving IL-33, IL-2 and IL-9 and different types of cells is pathogenically amplified in $Cftr^{-/-}$ mice. The finding that IL-33, IL-2 and IL-9 are also elevated in CF patients[33,34] prompted us to evaluate the contribution of cystic fibrosis transmembrane conductance regulator (CFTR) dysfunction on the activation of the inflammatory circuit. Given that CFTR dysfunction on both epithelial and myeloid cells impacts on lung inflammation[35], we assessed chimeric mice with CFTR unresponsive myeloid or epithelial cells for lung damage and inflammation, production of IL-33, IL-2 and IL-9 and MC/ILC2 activation upon *Aspergillus* infection. We found that epithelial cell damage and lung inflammation (Fig. 5a), levels of cytokines and IgE (Fig. 5b), MC and ILC2 expansion (Fig. 5c) were all attenuated or decreased in condition of CFTR deficiency in epithelial cells but CFTR sufficient myeloid cells, a finding suggesting that myeloid, and perhaps lymphoid, deficiency could contribute to the activation of the inflammatory circuit in CF. This seems to be the case as the opposite findings were observed in recipient C57BL/6 mice receiving CFTR deficient myeloid cells (Fig. 5a,c) These mice, however, showed an intermediate inflammatory phenotype as compared to $Cftr^{-/-}$ mice, a finding suggesting that, although to a different extent, CFTR deficiency on epithelial and myeloid cells may predispose to lung inflammation in response to microbial and non-microbial stimuli.

**The $IL9$ rs2069885 SNP correlates with high IgE levels in CF females**. To assess whether IL-9 may contribute to allergy in CF patients, we determined the effect of the non-synonymous $IL9$ p.Thr117Met (c.350C>T, rs2069885) polymorphism, known to be associated with lung function and sensitization[36,37] on total and *Aspergillus*-specific IgE levels in CF patients (Supplementary Table 1). Previous association studies demonstrated the existence of sex dimorphism linked to this polymorphism[36,37]. Therefore, we carried out association testing separately in males and females. The distribution of the total IgE was skewed but after natural logarithmic transformation, the distribution adequately approximated a normal distribution. $IL9$ rs2069885 genotype distribution did not deviate from Hardy–Weinberg equilibrium ($\chi^2$ test, $P = 0.776$) and it displayed a minor allele frequency of 0.129, comparable to the European population from 1000 Genome Consortium, minor allele frequency = 0.128 (ref. 38). $IL9$ rs2069885 genotype distribution did not differ between males and females ($\chi^2$ test, $P = 0.243$), but a significant rs2069885-sex interaction on total IgE levels was found (general linear model, $P = 0.004$), where female T allele carriers showed high IgE levels (linear regression: females, $\beta = 0.624$, $P = 0.043$; males,

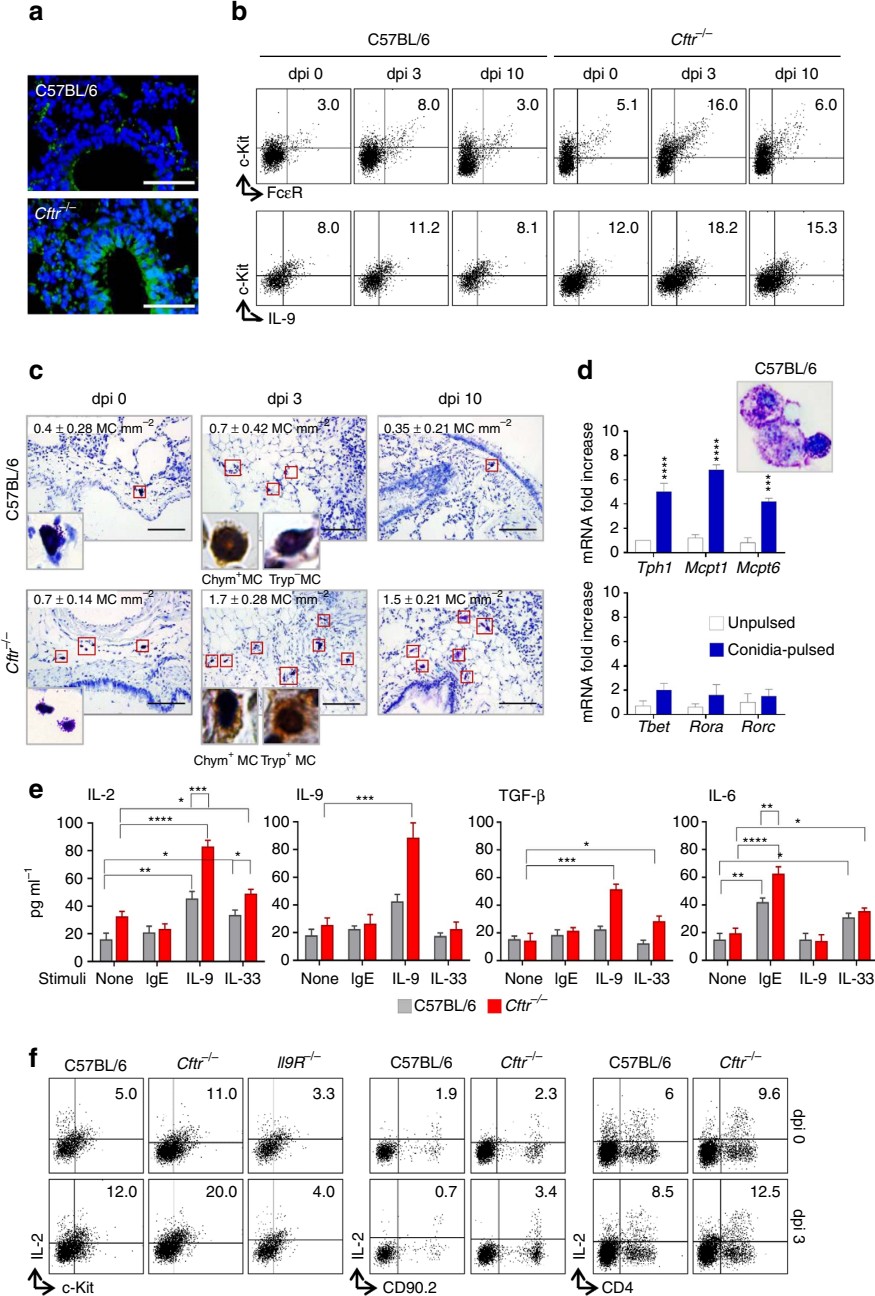

**Figure 3 | IL-9 activates mast cells to produce IL-2.** C57BL/6 or $Cftr^{-/-}$ mice (six per group) infected intranasally with live *A. fumigatus* conidia were evaluated at different days after infection (dpi) for (**a**) deposition of DNA on lung epithelial cells by TUNEL, resulting in bright DNA staining; (**b**) detection of c-Kit$^+$Fc$\varepsilon$R$^+$ and c-Kit$^+$IL-9$^+$ lung mast cells (MC) by flow cytometry (numbers refer to percentages of positive cells); (**c**) toluidine blue, relative MC number mm$^{-2}$ and, in the inset, immunohistochemical staining for chymase- and tryptase-positive MC in lung section. Photographs were taken with a high-resolution microscope (Olympus DP71) equipped with a $\times 40$ objective and (in the inset) a $\times 100$ objective and with EVOS FL Color Imaging System with a $\times 60$ objective (immunohistochemical staining). (**d**) Toluidine blue stain and transcription factors expression (PCR with reverse transcription) of c-Kit$^+$ cells magnetically isolated from lung of uninfected C57BL/6 mice and pulsed with live *A. fumigatus* conidia. (**e**) Cytokine production (mean values $\pm$ s.d., ELISA on culture supernatants) by purified lung c-Kit$^+$ cells, pulsed with *A. fumigatus* and stimulated with IgE, IL-9 and IL-33; (**f**) detection of c-Kit$^+$IL-2$^+$, CD90.2$^+$IL-2$^+$ and CD4$^+$IL-2$^+$ lung cells by flow cytometry (numbers refer to percentages of positive cells). *$P < 0.05$, **$P < 0.01$, ***$P < 0.001$, ****$P < 0.0001$, conidia-pulsed versus unpulsed c-Kit$^+$ cells, stimulated versus unstimulated c-Kit$^+$ cells and $Cftr^{-/-}$ versus C57BL/6 c-Kit$^+$ cells (data represent pooled results or representative images from three experiments, Two-way ANOVA, Bonferroni post test). *Mcpt1*, mast cell protease 1; *Mcpt6*, tryptase beta 2; *Rorc*, retinoic acid receptor–related orphan receptor C; *Rora*, RAR-related orphan receptor alpha; *Tbet*, T box expressed in T cells; *Tph1*, tryptophan hydroxylase 1.

$\beta = -0.666$, $P = 0.034$; Fig. 6a). In a subset of CF patients ($N = 114$; 57 males and 57 females), *Aspergillus*-specific IgE were also significantly higher in *IL9* rs2069885-T carriers (general linear model, $P = 0.002$; Fig. 6b). Notably, in line with the results

obtained on total IgE, the association of *IL9* rs2069885-T allele with higher *Aspergillus*-specific IgE levels was observed in females (linear regression, $\beta = 1.417$, $P = 0.002$) more than males (linear regression, $\beta = 0.489$, $P = 0.279$). We also tested other

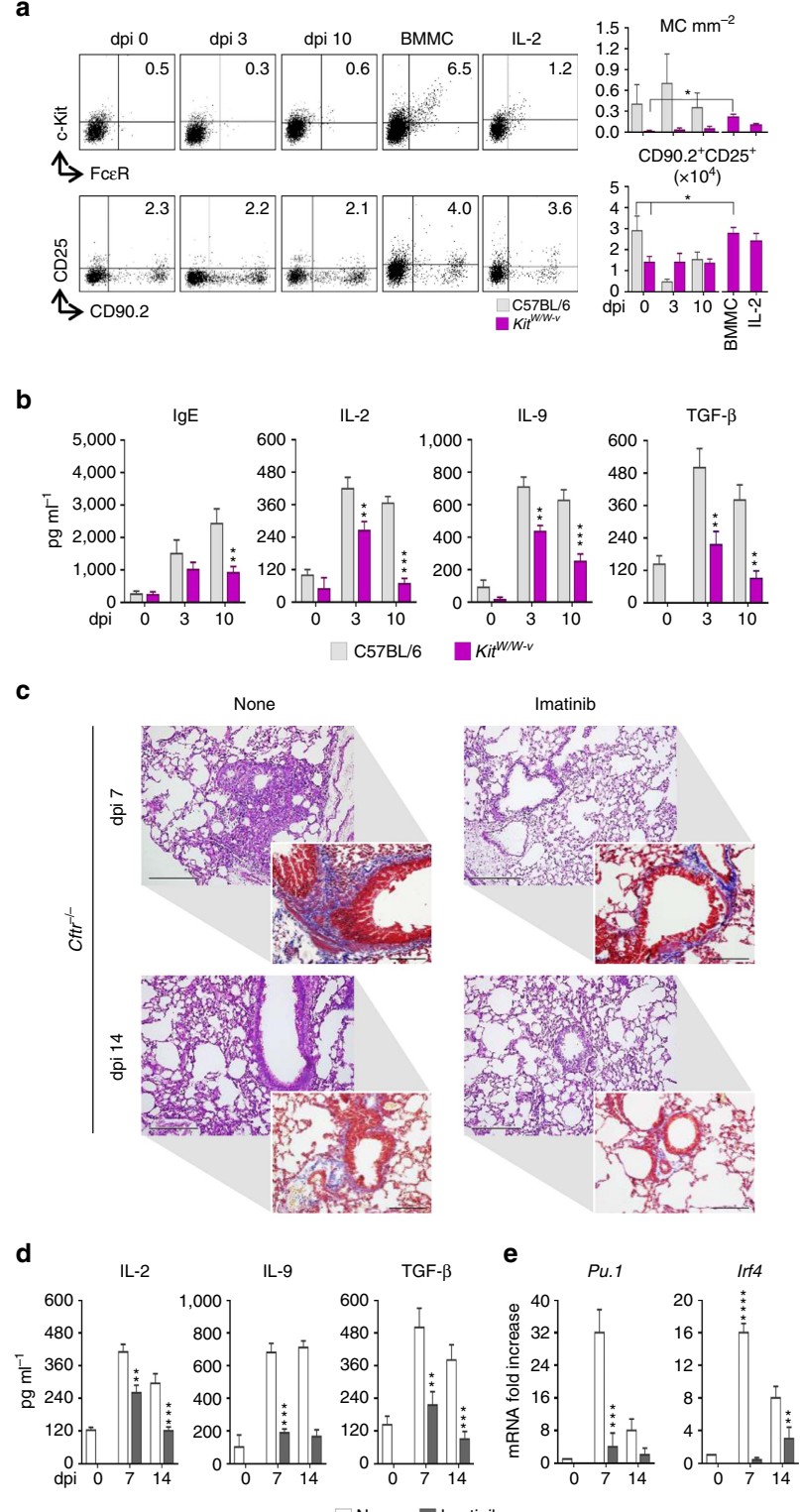

**Figure 4 | IL-9 activates mast cells to produce IL-2.** MC-deficient C57BL6-$Kit^{W/W-v}$ mice (six per group) were infected intranasally with live *A. fumigatus* conidia, engrafted intravenously with wild-type bone marrow-cultured mast cells (BMMC) or treated intraperitoneally with IL-2 for a week and assessed for (**a**) c-Kit$^+$Fc$\varepsilon$R$^+$ lung mast cells (MC) and CD90.2$^+$CD25$^+$ lung ILC2 by flow cytometry (numbers refer to percentages of positive cells) with relative cell number and (**b**) IgE and cytokine production. (**c**) Lung histology (periodic acid–Schiff and Masson's trichrome staining, in the insets); (**d**) cytokine production and (**e**) Th9-cell specific transcripts expression in $Cftr^{-/-}$ mice infected as above and treated with imatinib intraperitoneally for a week. Photographs were taken with a high-resolution microscope (Olympus DP71) equipped with a ×20 objective, scale bars, 200 μm and a ×40 objective (insets of **c**, scale bars, 100 μm). Results are mean values ± s.d., ELISA on lung homogenates for cytokines and PCR with reverse transcription on lung CD4$^+$T cells. *$P < 0.05$, **$P < 0.01$, ***$P < 0.001$, ****$P < 0.0001$, MC-deficient C57BL6-$Kit^{W/W-v}$ versus C57BL/6 mice or imatinib-treated versus untreated (none) mice (data represent pooled results or representative images from three experiments, Two-way ANOVA, Bonferroni post test). *Irf4* = interferon regulatory factor 4; *Pu.1* = purine-rich box 1.

*IL9* tagSNPs (rs2069882, rs31564, rs1859430, rs1799962; Supplementary Tables 2 and 3), encompassing the whole *IL9* gene region, but none associated with IgE levels. Linkage disequilibrium (LD) analyses of the five SNPs in *IL9* failed to reveal the presence of significant LD blocks (Supplementary Fig. 7). Haplotype analysis yielded no significant result

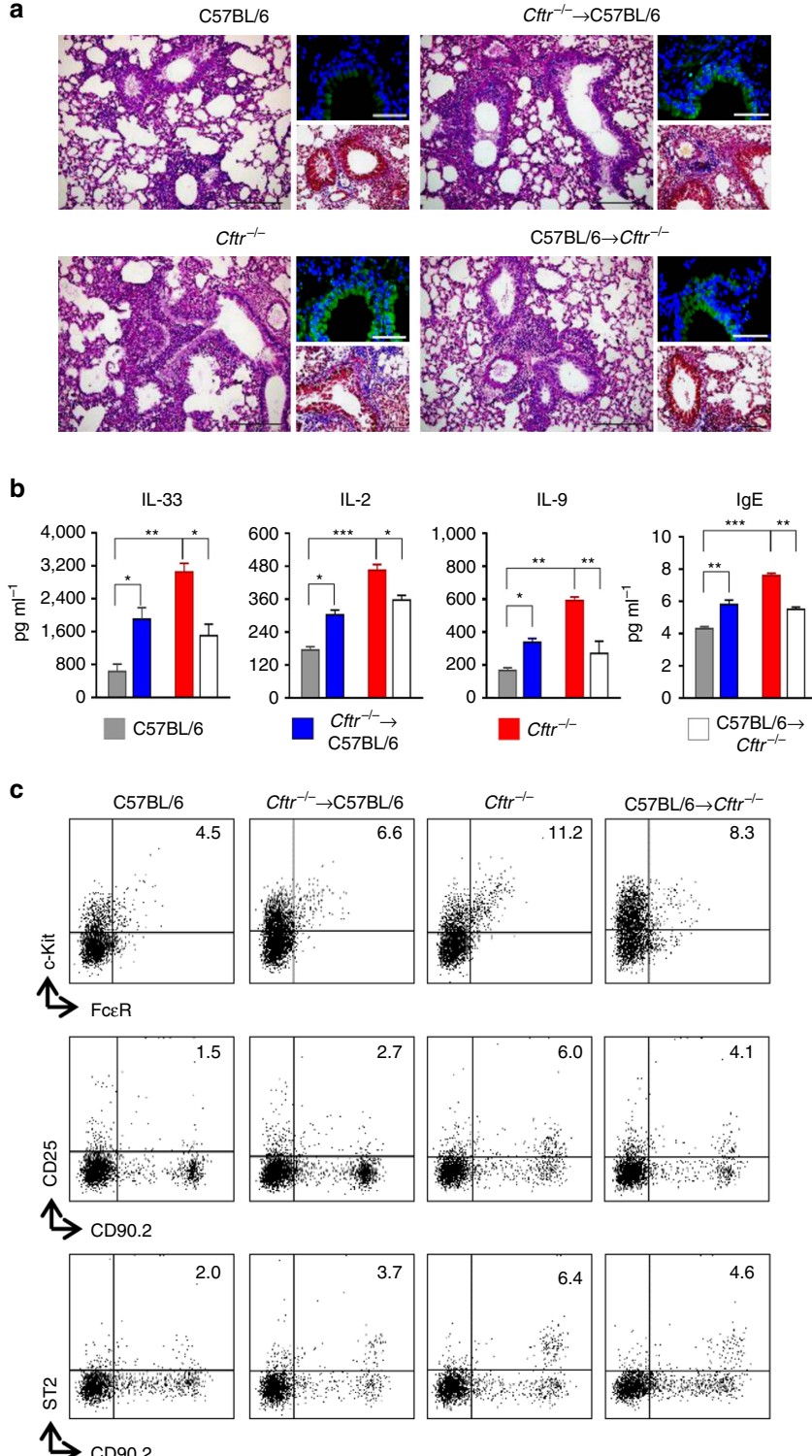

**Figure 5 | Epithelial and myeloid CFTR deficiency contribute to the inflammatory phenotype.** C57BL/6, *Cftr*[−/−] and chimeric C57BL/6 and *Cftr*[−/−] mice (10 per group) received $10 \times 10^6$ viable bone marrow cells 4 weeks before the intranasal infection with *A. fumigatus*. Chimeric mice were evaluated 7 days after the infection for (**a**) lung histology (periodic acid–Schiff and, in the insets, Masson's trichrome and TUNEL staining); (**b**) cytokines and IgE levels (mean values ± s.d., ELISA on lung homogenates); (**c**) detection of c-Kit[+]FcεR[+] mast cells, CD90.2[+]CD25[+] and CD90.2[+]ST2[+] type 2 ILCs by flow cytometry (numbers refer to percentages of positive cells in the lung). Photographs were taken with a high-resolution microscope (Olympus DP71) equipped with a ×20 objective; scale bars, 200 μm and a ×40 objective (insets of panel **a**, scale bars, 100 μm). \*$P < 0.05$, \*\*$P < 0.01$, \*\*\*$P < 0.001$, *Cftr*[−/−] versus C57BL/6, chimeric C57BL/6 versus C57BL/6, chimeric *Cftr*[−/−] versus *Cftr*[−/−] mice, Two-way ANOVA, Bonferroni post test.

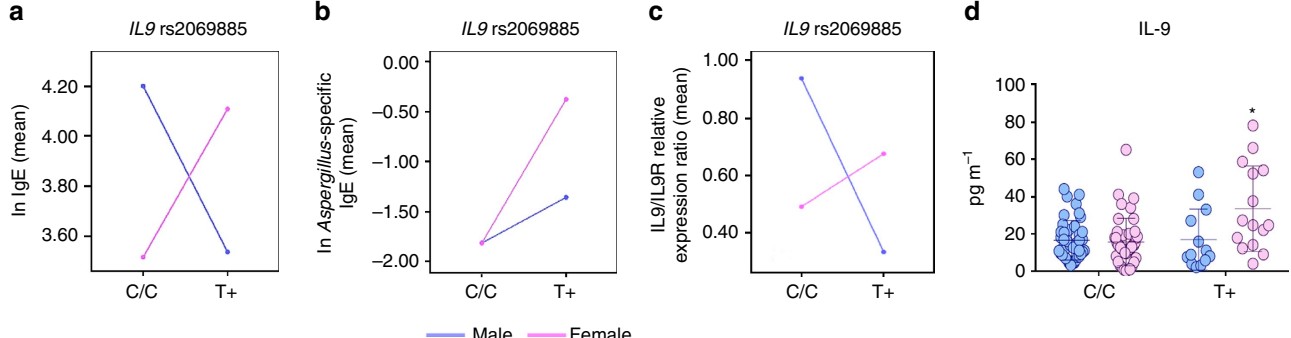

**Figure 6 | The *IL9* rs2069885 polymorphism correlates with high IgE levels in CF females.** (**a**,**b**) *IL9* rs2069885 sex interaction on total or *Aspergillus*-specific IgE levels and (**c**) *IL9/IL9R* expression ratio measured on CF patients. (**d**) Determination IL-9 (mean ± s.d., ELISA) in expectorates from CF patients carrying diverse genotypes at rs2069885.

(Supplementary Table 4), although the most common haplotype C–A–A–C–A (frequency 43.2%) turned out to be associated with higher IgE levels in males (linear regression, $\beta = 0.567$, $P = 0.022$).

We next determined whether the reported sex-specific associations could be attributable to a differential expression of *IL9* and its receptor, which is located on the pseudoautosomal region Xq/Yq, between males and females. A significant *IL9* rs2069885 by sex interaction was found when analysing the *IL9/IL9R* expression ratio (general linear model, $P = 0.025$) where T allele carriers showed opposite effects according to sex (linear regression, females, $\beta = 0.375$, $P = 0.116$; males, $\beta = -0.426$, $P = 0.375$; Fig. 6c) paralleling the finding on specific IgE (Fig. 6a) and on IL-9 levels in expectorates (Fig. 6d). No significant difference was noticed between males and females when *IL9* or *IL9R* gene expression levels were analysed independently.

Although these results are to be considered with caution given the small sample size–despite the sufficient power ($\beta = 0.20$, Power $= 0.8$; Supplementary Table 6)—which demands for a definitive validation using a larger new cohort of CF patients, the present findings confirm the existence of a sex dimorphism at *IL9* and *IL9R* loci, as already reported in several respiratory-related human phenotypes[36,37].

## Discussion

We have shown that IL-9 may expand pro-inflammatory CD25[+] ILC2/Th9 cell in CF, an activity involving the production of IL-2 by MC. Human MC co-localize near ILC2 in the human lung and could directly promote ILC2 responses *in vitro*[39]. We found here that, in addition to CD4[+] T cells, known to contribute to CD25[+] ILC2 expansion via IL-2 (ref. 8), lung MC may also affect the expansion of CD25[+] ILC2 thought to contribute to chronic inflammation through multiple mechanisms[3]. Failure to expand CD25[+] ILC2 occurred indeed in MC-deficient mice or mice treated with imatinib.

MC hyperplasia during chronic allergen challenge is associated with remodeling of airways[40]. In a mouse model of ovalbumin-induced airway inflammation, the influx of MC into lung peaks early after allergen challenge to mature over 14 days into cells expressing lower level of c-Kit, FcεRI and integrins[41]. MC hyperplasia was observed in the lung of *Cftr*[−/−] mice along with the detection of tryptase-positive and chymase-positive MC, known to be expanded in asthmatic patients[42] and in disease areas of CF lung[31]. Tryptases and chymases contribute to inflammation and tissue remodeling through the selective proteolysis of matrix proteins and the activation of protease-activated receptors and matrix metalloproteinases[29]. Consistent with the finding that MC from CF patients are not high in FcεR1

expressing[43], we found that lung MC from these mice poorly responded to IgE in terms of IL-6 production but released IL-2, in addition to TGF-β, in response to IL-9.

IL-9 is a pleiotropic cytokine that has multiple effects on structural as well as numerous hematopoietic cells, which are central to the pathogenesis of asthma[44,45]. The important role for the IL-9-MC axis in the pathology associated with chronic allergic inflammation has been already described[46]. IL-9 not only stimulates MC growth and expansion but also stimulates changes in gene expression that might alter responsiveness to other stimuli[47]. *In vivo*, IL-9 governed allergen-induced MC numbers in the lung, and anti-IL-9 antibody-treatment protected from airway remodeling, decreased expression of the profibrotic mediators TGF-β and improved lung function[48]. The correlation between a reduction in MC numbers and decreased airway remodeling, after IL-9 inhibition, is consistent with reports that MC-deficient mice demonstrate significantly attenuated fibrosis and inflammation after silica[49], ozone[50], or bleomycin injury[51].

We found that not only were IL-9 production and MC expansion significantly increased in CF mice but that the IL-9-MC axis contributed to the expansion of CD25[+] ILC2 leading to Th9 cell activation that further contributed to the allergic inflammatory pathology (Fig. 7). IL-9 promoted IL-2 production by lung MC from *Cftr*[−/−] mice, a finding that may explain the increased and persistent expansion of CD25[+] ILC2 in these mice. CD25[+] ILC2 were indeed not expanded in condition of IL-9 ablation or MC-deficiency, a finding suggesting that the IL-9/MC/IL-2 axis drives the expansion of CD25[+] ILC2. In addition, as IL-2 stimulated lung CD25[+] ILC2 to produce IL-9 (ref. 9), this may have a positive feedback effect on ILCs, since lung ILCs cultured with IL-9 increased the production of type 2 cytokines[9] and up-regulated the anti-apoptotic protein BCL-3, thereby promoting ILC2 survival[8]. Of great interest, Lin[−] cells from CF lung also promoted Th9 cell activation *in vitro*, an activity that required IL-9R expression on responding CD4[+] T cells. Thus, the pathogenic role of IL-9 in promoting allergic inflammation may go beyond CD25[+] ILC2 expansion to include the activation of Th9 cells. That Th9 cells are a major source of IL-9 in models of allergic inflammation and play an important role in MC accumulation and activation has been reported[52]. By producing IL-9, Th9 cells may in turn serve as a positive loop amplifying the IL-9/MC/ILC2 axis, promoting a deleterious vicious circle in which the production of profibrotic TGF-β by IL-9-stimulated MC plays a plausible important role. The presence of TGF-β dependent signalling in areas of prominent fibrosis in CF has been already documented[24] along with its inhibition of chloride channel activities[53] and the association of the TGF-β genetic variants with more severe lung disease[54].

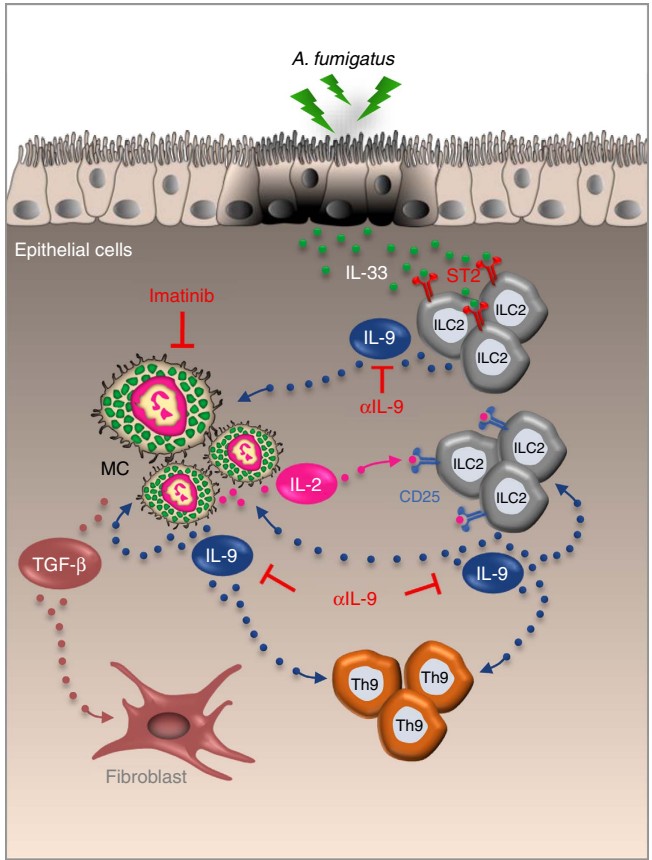

**Figure 7 | Proposed model for the role of IL-9 in promoting a mast cells/ILC2/Th9 fibrotic pathway in CF.** IL-9, produced by IL-33-expanded ILC2, activates MC for IL-2 production leading to the expansion of CD25$^+$ILC2 that promote Th9 cell activation. The resulting increased production of IL-9 further amplifies the inflammatory loop by promoting ILC2 survival and type 2 cytokines production and by activating MC for the production of fibrotic TGF-β. IL-9 ablation or MC inhibition (imatinib) are potential drugable pathways through which inflammation and allergy could be restrained in CF. EC, epithelial cells. αIL-9, IL-9 neutralizing antibody.

Of note, together with IL-4, is able to promote Th9 cell development *in vitro*[55], a finding highlighting the potential role of TGF-β in reinforcing Th9 activity *in vivo*. However, TGF-β production by MC in response to IL-9 may also serve a regulatory, anti-inflammatory role[11,56].

In addition to IL-9, IL-33 is also a crucial regulator of MC functions and both IL-33 and MC have been influentially associated to the pathophysiology of allergic diseases and inflammation[57]. IL-33 is expressed in epithelia from patients with CF and potentiates neutrophil recruitment[58] as well as in type-2 pneumocytes on allergic lung inflammation[59]. In this study, IL-33 was increased in CF mice and likely correlated with the expansion of ST2$^+$ILC2. At variance with IL-2, IL-33 did not stimulate MC for IL-9/TGF-β production, a finding indicating a minor contribution of the IL-33/MC axis in promoting inflammatory allergy and pathology in response to the fungus. As a matter of fact, the IL-33/MC/IL-2 axis was found to suppress, rather than promote, papain-induced allergic inflammation by promoting Treg[7].

Airway inflammation and recurrent pulmonary infections play a central role in the progression of CF lung disease. It is still an open question whether CFTR deficiency *per se* may enhance the inflammatory response to the different environmental cues[60]. Our data would suggest that CFTR dysfunction on both epithelial and myeloid cells may impact on the inflammatory circuit leading to the activation of the inflammatory IL-9/Th9 pathway: namely, among others[35], on both epithelial damage eventually leading to overproduction of IL-33 and on the MC propensity to respond to IL-9 with IL-2. However, PU.1 translocation into the nucleus has also been shown to be significantly higher in CF monocytes than in controls[61]. It seems that defective CFTR impacts on the mechanisms of communication between innate and adaptive immune response.

The potential contribution of IL-9 to CF pathogenesis in humans is unknown. Elevated levels of IL-9 were observed in the expectorates from CF patients, likely accounting for the expansion of TGF-β-producing MC in diseased lung areas[31]. It is of interest that in human asthmatic lung tissue, MC were the main IL-9R expressing population[46]. It seems therefore that the IL-9/MC/IL-2 axis may have a pathogenic role in CF patients and that its targeting could lead to a reduction in chronic inflammation and improved lung function of these patients. Studies have indeed highlighted the importance of Th9 cells in allergic lung inflammation by promoting epithelial alterations, goblet-cell hyperplasia, mucus production and infiltration of MC and eosinophils[46]. Considering that the costimulatory signal OX40 is required for Th9 activation[62], it is intriguing that OX40 ligand was critical in driving Th2-allergic responses to *A. fumigatus* in peripheral CD4$^+$ T cells isolated from CF patients with ABPA (ref. 63). It is clear that the inflammatory response in the lung involves different Th cell types whose specific role in CF remain unclear. Intriguingly, there are conditions where promoting IL-9 might be therapeutically beneficial[11] and ILC2 and MC could be exploited for pathogen immunity and tissue repair[64]. In this regard, the fact that imatinib apparently exacerbated signs of inflammation during infection points to a some beneficial role c-Kit$^+$ cells may have in the control of *Aspergillus* infection. MC indeed exhibited conidiocidal activity (Supplementary Fig. 6d), a finding suggesting that MC could serve as tissue sentinels modulating antifungal immune responses, as suggested[65].

In conclusion, IL-9 and MC may have an important role in the pathogenesis of lung disease and inflammation in CF. Considering the inherent resistance to steroids of MC in asthmatic patients[66], a better understanding of cellular and molecular pathways leading to inflammation and impaired lung functions may inspire new treatment avenues in patients with CF. Our study would suggest that imatinib, known to inhibit lung fibrosis[67], could be therapeutically exploited in CF patients with an exalted IL-9/Th9 responses. In addition, it is of great interest that the *IL9* rs2069885 polymorphism, linked to high IgE levels, was associated with females more than males with CF, a finding offering an explanation for the, as yet unexplained, 'gender gap' in mortality between females and males in CF[68] and fostering gender medicine in CF.

## Methods

**General experimental approaches.** Mice were randomized and assigned to group allocation at the time of purchase to minimize any potential bias. No blinding was applied on harvesting cells after the treatments.

**Mice.** C57BL/6 (wild-type, WT), *Rag1*$^{-/-}$ and MC-deficient C57BL/6-*Kit*$^{W/W-v}$ mice, 6–8 week old, of both sexes, were purchased from Charles River (Calco, Italy). Genetically engineered homozygous *Cftr*$^{-/-}$ mice[69] were bred at the CF core animal facility at San Raffaele Hospital, Milan, Italy. *Il9R*$^{-/-}$ and *RagIl9R*$^{-/-}$ mice were from the Ludwing Institute for Cancer Research, Brussells.

**Fungal infection allergy and treatments.** Anaesthetized (by inhalation of 3% isoflurane (Forane Abbot) in oxygen)) mice were infected by the intranasal instillation of $2 \times 10^7$ resting conidia/20 μl saline. For allergic broncho-pulmonary aspergillosis, *A. fumigatus* culture filtrate extract in incomplete Freund's adjuvant

(Sigma-Aldrich) was given (100 μg) to intact mice intraperitoneally (i.p.), subcutaneously and then intransally (20 μg), twice a week apart. A week after the last intranasal challenge, mice received $10^7$ Aspergillus resting conidia and evaluated a week later. Murine monoclonal anti-IL-9 antibody (MM9CI from BioXcell), or control isotype IgG, were administered i.p. at the dose of 500 μg kg$^{-1}$ for a week starting the day of the infection. The levels of IL-9 after antibody treatment were $65 \pm 15$ versus $295 \pm 16$ pg mg$^{-1}$ for C57BL/6 and $103 \pm 27$ versus $719 \pm 14$ pg mg$^{-1}$ for Cftr$^{-/-}$ mice, treated versus untreated mice. IL-2 at the dose of 1 μg per mouse was given i.p. for a week. Imatinib mesylate (Glivec, ST1571 Novartis, Basel) were administered i.p. at the dose of 1 mg kg$^{-1}$ for a week starting the day of the infection.

**Mast cell engraftment.** Selective engraftment of MC in MC-deficient C57BL/6-Kit$^{W/W-v}$ mice was performed as follow. Briefly, bone marrow cells derived from 6-week-old female C57BL/6 mice were cultured in WEHI-3–conditioned medium (ATCC number TIB-68), as a source of IL-3, for 4–5 weeks to obtain MC populations (BMCMC) which purity was higher than 95%. Via the tail vein, $5 \times 10^6$ BMCMC were injected into each mouse, and the recipients were used for experiments 4 weeks later.

**Generation of bone marrow chimeras.** Femurs and tibias were removed aseptically from donor C57BL/6 and Cftr$^{-/-}$ euthanized mice. Bone marrow was retrieved by flushing with cold Dulbecco's modified Eagle's medium supplemented with 10% heat-inactivated foetal calf serum and 2 mM L-glutamine (Invitrogen). Cells were washed twice with PBS without calcium and magnesium supplemented with 1% foetal calf serum. Recipient C57BL/6 and Cftr$^{-/-}$ mice were irradiated with 9 Gy and reconstituted no later than 6 h after the last irradiation with $10 \times 10^6$ T cells by intravenous injection. Mice were given sulfamethoxazole (150 mg ml$^{-1}$) and trimethoprim (30 mg ml$^{-1}$) in drinking water for the first 3 weeks of reconstitution. Mice were used no earlier than 4 weeks after transplantation. Before use in experiments, all mice were bled from the retro-orbital plexus, and the peripheral blood lymphocytes were analysed for the stable donor-type chimerism by reverse transcription-PCR of Cftr.

***In vivo* staining analysis.** For histology, paraffin-embedded tissues were stained with Periodic acid-Schiff, Masson's trichrome or Toluidine Blue staining to investigate inflammation, collagen deposition and MC infiltration, respectively. For immunofluorescence, lungs were incubated at 4 °C with phycoerythrin-conjugated (PE) anti-CD25 (Miltenyi Biotec clone 7D4, 1:60), anti-T1-ST2 (BioLegend clone DIH9, 1:400), anti-IL-9 (Milenyi Biotec clone RM9A4, 1:60) and fluorescein isothiocyanate-conjugated (FITC) anti-mouse CD90.2 (Miltenyi Biotecclone 30-H12, 1:60) and anti-CD4 (BioLegend clone GK1.5, 1:1,000). Nuclei were counterstained with 4,6-diamidino-2-phenylindole. Immunostaining with appropriate irrelevant antibodies did not give positive staining of the lung. For immunohistochemistry, the lung sections were incubated overnight with polyclonal anti-chymase (Bioss, 1:100) or monoclonal anti-tryptase (Abcam clone EPR8476, 1:500) followed by the secondary biotinylated antibodies. Cells were counterstained with haematoxylin. Photographs were taken using a high-resolution Olympus DP71 microscope with a $\times 20$ and $\times 40$ objective or EVOS FL Color Imaging System with a $\times 60$ objective. For immunoblotting, blots of lung lysates were incubated with polyclonal antibodies against STAT5 and phospho-STAT5 (both from Cell Signaling, 1:1,000) and normalized on β-actin (clone AC-15 from Sigma). The ChemiDocTM XRS + Imaging system (Bio-Rad) was used to detect chemiluminescence on the addition of the LiteAblotPlus chemiluminescence substrate (Euroclone S.p.A). Quantification was done by densitometry image analysis using Image Lab 5.1 software (Bio-Rad). The uncut blot is shown in Supplementary Fig. 8.

**TUNEL assay of lung sections.** Sections of lungs fixed in 4% buffered paraformaldehyde, pH 7.3, for 36 h and embedded in paraffin, were deparaffinized, rehydrated, treated with 0.1 M citrate buffer, pH 6.0, washed and blocked in 0.1 M Tris-HCl buffer, pH 7.5, supplemented with 3% bovine serum albumin and 20% foetal calf serum. The slides were then incubated with fluorescein-coupled dUTP and terminal deoxynucleotidyltransferase–mediated deoxyuridine triphosphate nick-end labelling (TUNEL) enzyme (Roche Diagnostics) in the presence of terminal deoxynucleotidyltransferase. Unspecific binding was removed by washing with phosphate-buffered saline for 10 min at 70 °C. The sections were mounted and analysed by fluorescence microscopy, using a $\times 40$ objective.

**Cell isolation and culture.** Lungs were finely minced, digested in 16 mg ml$^{-1}$ Collagenase P (Roche) for 30 min and meshed through a 70-μm cell strainer. ILCs were isolated from total lung cells by magnetic depletion of Lineage Positive cells (Miltenyi Biotec). CD4$^+$ T cells and c-Kit$^+$ cells were purified from total lung cells after incubation of CD4 microbeads and with PE-labelled anti-c-Kit followed by anti-PE MicroBeads respectively (both from Miltenyi Biotec). FcεRIα-APC and c-Kit-PE (Miltenyi Biotec) staining and morphological examination after toluidine blue staining on the cytospin slides were used fir c-Kit$^+$ cell phenotyping. For Lin$^-$-CD4$^+$ T cell co-culture, $2 \times 10^6$ CD4$^+$ T cells were co-cultured with $1 \times 10^6$ Lin$^-$ cells with or without 50 ng ml$^{-1}$ recombinant IL-33, 40 ng ml$^{-1}$

recombinant IL-2 and pulsed with *A. fumigatus* conidia. Three days later, IL-9 levels in culture supernatants were analysed by ELISA and Th9 transcription factors by real-time PCR. To separate Lin$^-$ and CD4$^+$ T cell, we used the Transwell culture system (Costar, 0.4 μm pore size; Corning) with CD4$^+$ T cells in the lower wells and Lin$^-$ in the upper wells[7]. For c-Kit$^+$ cells culture, $5 \times 10^5$ cells were cultured overnight in RPMI medium and pulsed with *A. fumigatus* conidia with or without 10 μg ml$^{-1}$ IgE, 100 ng ml$^{-1}$ IL-33 and 100 ng ml$^{-1}$ IL-9.

**Flow cytometry.** Flow cytometry on enriched Lin$^-$ cells was performed with a combination of the following fluorescence-conjugated mAbs (all from Miltenyi Biotec unless specified otherwise): APC-conjugated anti-NKp46 (29A1.4.9), anti-CD90.2 (30-H12), anti-Rorγ (t; REA278), anti-FcRIa (MAR-1), anti-CD4 (GK1.5); PE-conjugated anti-NK1.1 (PK136), anti-CD25 (7D4), anti-T1-ST2 (DIH9, from Biolegend), anti-CD117 (3C11), anti-IL-9 (RM9A4) and anti-IL-2 (JES6-5H4). For intracellular staining, phorbol 12-myristate 13-acetate (PMA)/ionomycin-stimulated cells were added of brefeldin, and then permeabilized with the CytoFix/CytoPerm kit (BD Biosciences) for intra-cytoplasmic detection of IL-9 and IL-2. Flow cytometry was done at 4 °C on cells first exposed to Fc receptor mAb (2.4G2). Cells were analysed with a BD LSRFortessa flow cytometer equipped with BD FACSDiva 7.0 software.

**ELISA and real-time PCR.** The levels of cytokines and IgE in lung homogenates, culture supernatants or expectorates were determined by ELISA kits (R&D Systems) following manufacturer's instructions. Real-time PCR with reverse transcription was performed using CFX96 Touch Real-Time PCR Detection System and SYBR Green chemistry (Bio-Rad) on total RNA reverse transcribed with the cDNA Synthesis Kit (Bio-Rad). The PCR primers were as listed in Supplementary Table 5. Amplification efficiencies were validated and normalized against *Gapdh*. The thermal profile for SYBR Green real-time PCR was at 95 °C for 3 min, followed by 40 cycles of denaturation for 30 s at 95 °C and an annealing/extension step of 30 s at 60 °C. The messenger RNA-normalized data were expressed as relative gene messenger RNA in treated versus untreated groups or cells.

**Human study.** A cohort of 347 patients of Caucasian origin with a proven diagnosis of CF (CFTR genotyping, sweat testing and clinical phenotype) was enroled in a prospective multicenter longitudinal genetic association study, See Supplementary Table 1 for clinical data including age, gender, lung function testing, measures of nutrition, microbiological findings and vital status of the patients' cohort.

**SNPs selection and genotyping.** DNA was isolated from blood with the QIAamp DNA Mini (Qiagen, Milan, Italy) system and stored at $-20$ °C. IL9 SNPs were selected based on literature review[37] and their ability to tag surrounding variants in the HapMap-CEU population of the International HapMap project, NCBI build B36 assembly HapMap phase III (http://www.hapmap.org). Haplotype-based tagging SNPs were selected by assessing LD blocks from the genes of interest with a pairwise correlation coefficient $r^2$ of at least 0.80 and a minor allele frequency higher than 5% in the HapMap-CEU population. Five IL9 SNPs complied with the selection criteria: rs2069885, rs2069882, rs31564, rs1859430 and rs1799962. The applied Biosystems 7500 Fast qPCR system (Life Technologies) was used for SNP genotyping by KASPar assays (KBiosciences, Hertfordshire, UK). Each genotyping set comprised randomly selected replicates of sequenced samples and negative controls. Agreement between original and duplicate samples was $\geq 99\%$ for all SNPs. Laboratory personnel were blind to the sample status.

**Statistical analysis.** Data are expressed as mean $\pm$ s.d. Horizontal bars indicate the means. Statistical significance was calculated by two-way ANOVA (Bonferroni's *post hoc* test) for multiple comparisons and by a two tailed Student's *t*-test for single comparison. The distribution of levels tested by Kolmogorov–Smirnov normality test turned out to be non-significant. Values of *P* not > 0.05 were considered significant. The data reported are either representative from two or three experiments (FACS data, histology, immunofluoresce and TUNEL assay) or pooled otherwise. The *in vivo* groups consisted of 6 mice per group. Data were analysed by GraphPad Prism 4.03 programme (GraphPad Software). No statistical method was used to predetermine sample size. Genetic association testing was carried out considering additive and dominant models by linear regression implemented in Plink v1.07 (ref. 70), adjusting for age at sampling. Haplotype-based association tests were performed by general linear model using Plink v1.07 (ref. 70). LD analysis was performed using Haploview, and defining LD blocks based on the solid spine of LD algorithm. Analyses were conducted stratifying the study population according to sex since previous evidence highlighted the existence of sex dimorphism at IL9 locus[36,37]. IL9 rs2069885 by sex interaction was tested by general linear model using SPSS v.21. Two-tail *P* values are reported. Bonferroni's correction for multiple testing was not performed since we are assessing specific questions on a candidate gene and we are not searching for associations without a priori hypotheses. Power calculation (using QUANTO v1.2.4) was performed to determine whether the sample study had sufficient power to detect a significant

association between *IL9* rs2069885 and total IgE levels. Considering sample size ($N = 237$), allele frequency (13%), dominant genetic model, mean level of IgE and its standard deviation ($131.7 \pm 248.2$ U ml$^{-1}$) and $\alpha = 0.05$ two-sided, we could establish that our sample have sufficient power ($\beta = 0.20$, Power $= 0.8$) to detect an association explaining more than 3% of total IgE levels variance (Supplementary Table 6).

**Study approval.** All animal experiments were approved by local government authorities and were in agreement with the Italian Approved Animal Welfare Authorization 360/2015-PR and Legislative decree 26/2014 obtained from the Italian Ministry of Health lasting for five years (2015–2020). Infections were performed under anaesthesia and all efforts were made to minimize suffering. Human studies approval was obtained from institutional review boards the Bambino Gesù Children's Hospital (Rome, Italy), Ospedale Maggiore Policlinico, University of Milan, (Milan, Italy), Innsbruck Medical University, (Innsbruck, Austria) and Servizio di Supporto Fibrosi Cistica, (Cerignola, Foggia, Italy). Written informed consent was obtained from the participants, or, in case of minors, from parents or guardian.

**Data availability.** The data that support the findings of this study are available from the corresponding author upon request.

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

## Acknowledgements

The Specific Targeted Research Project FunMeta (ERC-2011-AdG-293714 to L.R.) supported this study. We thank the Italian Cystic Fibrosis Research Foundation for intellectual and technical support. Dr Marilena Pariano and Dr Matteo Puccetti have been recipient of a fellowship from the Italian Cystic Fibrosis Research Foundation.

## Author contributions

G.R. and V.O. performed *in vivo* murine experiments. C.G. and V.N. performed human genotyping. M.P., M.P. (Puccetti), R.G.I. and M.B. performed immunohistochemistry, immunofluorescence and histopathology; M.D.Z., G.P., T.Z. and C.E.P. contributed to mast cell experiments; J.-C.R. provided scientific advice; O.B. and P.S. provided reagents and contributed to experimental design; V.L., C.C., M.C.R., E.F., C.L.-F., F.M., G.R., H.E. and L.R. (Ratclif) provided the clinical samples. S.M., V.N.T., V.N. and L.R. designed the experiments, analysed data and wrote the paper.

## Additional information

**Competing financial interests:** The authors declare no competing financial interests.

