## [Peer Review File · Nature Communications]

Reviewer #1 (Remarks to the Author):

This is a topical and interesting area of study, as Th9 cells are a relatively a new subset, with many outstanding questions to be addressed. Here, Moretti et al. reported an interesting positive feedback loop amongst mast cells, ILC2 and Th9 cells to explain sustained IL-9 production in patients with cystic fibrosis. As presented, there are significant concerns with this paper, and some of the experiments are questionable and data may not their conclusions.

Major concerns:

1. The authors stated that the late IL-9 production is T cell-dependent (Line 98-99), but experiments demonstrating this claim are incomplete. At the very least, FACS plots showing lung CD4 T cells or Lin- ILC2 that produce IL-9 are important, simply using non-Th9 specific transcriptional factors such as PU.1 and IRF4, or IL-9 production in a pool of lung cells are insufficient to define Th9 cells. Furthermore, mRNA assessment in a pool of lung cells does not truly define specific functions of a subset of Th cells.
2. The manuscript would be more informative if the authors could provide absolute numbers of Lung ILC2 cells (Figure 1b), instead of only percentage, which can be misleading. The authors mentioned many times that CD25+ ILC2 expansion in *Cftr*^{-/-} mice, but CD25 is expressed by many other cell types also.
3. Upon *Aspergillus fumigatus* infection, the percentage of ILC2 is decreased dramatically (Fig 1b), while the transcriptional factor RoRa, which is believed to ILC2, is increased in WT C57BL/6 mice (Fig 1c). This paradox is not explained in the paper.
4. In Figure 2C, the immune staining of the lung sections by CD90.2 and CD25 or ST2 is insufficient to define ILC2 cells. Activated T cells, even Tregs, are CD25+, and should be stained also. However, IL9R KO lung sections are completely negative for CD25 staining, especially in mice infected with *Aspergillus fumigatus* where T cell activation is expected.
5. In Figure 2h, several key controls are missing. The IL-2 and IL-33 only groups should be included to rule out any direct effect of IL-2 and IL-33 on CD4 T cells.
6. In Figure 3e, in the ELISA assays performed for IL-9 production in cultures where exogenous IL-9 (100ng/ml) was added, how can you distinguish IL-9 produced from IL-9 added?

7. The authors state that MCs contribute to IL-2 production eventually leading to CD25+ ILC2 expansion (Line 169-170). But the experiments presented by no means show that IL-2 is from MC, not from activated T cells during infection. Does exogenous IL-2 rescue CD25+ ILC2 expansion in Figure 4a (or Line 175-176)?

8. Although the authors found a significant association between T allele of IL9 gene and IgE production in females, the patient number is very small, and caution should be used in making definitive conclusions.

Reviewer #2 (Remarks to the Author):

Moretti et al describe a mast cell/ILC2/Th9 circuit that regulates inflammation during fungal infection in wild type and CFTR-mutant mice. The authors show increased IL-9, ILC2, mast cell, and Th9 expansion during *A. fumigatus* infection, and that these responses are exaggerated in the *Cftr*^{-/-} mice. They show that IL-9/IL-9R is required for this response, and that IL-2 enhances Th9 cells co-cultured with ILCs and indicated by expression of IL-9, IRF4 and PU.1. They show that IL-9 stimulates IL-2 production from mast cells, and that the response is significantly increased in the absence of *Cftr*. Mast cells are required for this circuit because *Kit*^{-Wsh} mice and imatinib-treated mice have lower cytokine production and diminished Th9 gene expression. Finally the authors show sex specific effects of an IL9 polymorphism on IgE and IL-9 serum concentrations in CF patients.

This is a novel and interesting report. The authors have used a variety of approaches for defining this cellular circuit in models of fungal infection in wild type and CFTR-mutant mice. The inclusion of correlations in data from patients adds to the depth of the study. However, there are conclusions made that are not entirely supported, and some assumptions made that are not necessarily valid. The following are needed to complete this story.

1. It is presumed that when the authors examine PU.1 and IRF4 expression they are performing these assays in purified CD4 T cells, as both of these genes can be expressed in many other cell types. It would be important to show that these cells are actually making or capable of making IL-9. Minimally this could be done with the mRNA samples to examine IL9 expression. Optimally, seeing CD4+ T cells that are IL-9+ by flow cytometry would be more convincing.

2. The authors show that mast cells when stimulated ex vivo can make IL-2 when stimulated with IL-9, but it is not clear this is occurring in vivo. The authors need to show flow cytometry of cells directly ex vivo (unstimulated or PMA/ionomycin) that are stained for IL-2 and then show how much of the IL-2 is coming from mast cells versus other cell types. They further need to show this is lacking in anti-IL-9 treated mice or *Il9r-/-* mice, and increased in the *Cftr*-mutant mice. Evidence for this link in the circuit is lacking.

3. In a similar point, the authors are not clearly showing the relative contribution of IL-9 from the various cell types. The authors seem to assume ILC2 cells are making IL-9 (Fig. 6), but this is never shown. Both CD4 T cells and mast cells are shown to make IL-9, but in separate assays where it's hard to compare production. At least at the level of mRNA (and optimally with flow cytometry) the authors need to provide a comparison of IL-9 production from the 3 cell types, probably at several time points, so a reader can appreciate what each is contributing to the environment.

4. One point that the authors never address either experimentally or in the discussion is where the CFTR is having an effect. Is it in the cells defined within this circuit or in the lung cells that lead to different responses from the cell types. Obviously this could be a whole new area, but I wonder if a reciprocal bone marrow chimera approach to recapitulate data in Fig. 1 would at least determine if CFTR was affecting this circuit from hematopoietic cells versus airway or other structural cell.

5. It is not clear that the authors have purified ILCs when they examine *Rora*. As this gene is expressed in many cell types, this is critical. In general, the figure legends need to be clearer on what cells are being examined in assays for gene expression and cytokine production.

6. In Fig. 4a, the authors need to show, at least by graph, a comparison of the numbers of mast cells and ILCs between the B6 and *Kit-Wsh* mice. As shown, the data in that panel is hard to interpret.

7. The title needs to be modified. The data from patient samples is strength, but it does not fully support the circuit, nor does it support a statement that this circuit impacts pathology. Similarly, the mouse model, while having the CF mutation, does not develop the same pathology as patients and the pathology being examined in this report is from acute fungal infection. The authors can decide how to handle this, but at least should include the phrase '....lung pathology in a mouse model of cystic fibrosis'.

8. The writing needs to be edited by a native English speaker; there were some parts that were hard to read and understand.

Reviewer #3 (Remarks to the Author):

The paper by Moretti shows novel data on the role of IL-9 in influencing ILC2 and mast cell accumulation in response to Aspergillosis exposure - the data of which appears to have relevance in fungal sensitization in CF. The paper is well written and the data are novel and largely support the conclusions. There are a few issues with controls that need to be clarified.

1. In Figure 1, the fungal burden needs to be shown as the increased pathology in the CFTR^{-/-} mice could be due to greater antigen retention. Also total and fungal specific IgE responses, which is the hallmark of ABPA should be shown.
2. It is unclear what the authors mean between infected vs ABPA in Figure 2b. This needs to be clarified. Again IgE levels should be shown here.
3. Figure 3 should also include IgE. Can eth sash mice make IgE but not just activate mast cells?
4. Did the authors assess Aspergillus specific IgE in the clinical cohort?

Reviewer #4 (Remarks to the Author):

"A mast cell/ILC2/Th9 pathway promotes lung pathology in cystic fibrosis" by Moretti and colleagues examines the contribution of several effector mechanisms typically studied in allergic airway disease in the setting of cystic fibrosis. The key approach here is the use of the cftr^{-/-} mouse strain and studies on how these mice differ from controls in response to Aspergillus fumigatus infection. The authors reach the conclusion that there is a novel pathway through which mast cells coordinate with ILC2 cells to drive Th9 cell responses and pathogenesis. Importantly, the authors close the manuscript by showing associations of SNPs in the IL-9 locus with CF, which were rather intriguingly gender dependent.

In general, this study does provide aspects of data to support the fairly lengthy pathway that involves the three key cell types (mast cells, ILC2 and Th9) and the two key mediators (IL-2 and IL-9) but fails in the depth of inquiry to adequately determine if their conclusions are correct. The work also struggles in several places where there are points of the pathway that are not explained and/or the data seems to refute the mechanism to some degree. Consequently, the overall impression is that the work is an intriguing story but preliminary at this point.

Key specific areas of concern are as follows:

- 1) A concern that has impact on much of the data shown relates to the basal phenotype differences between the WT and the cftr^{-/-} strain. In several key figures, it is clear that there is a basal increase in many ILC2/Th2 cytokines and transcription factors (Fig 1C-F) and mast cells (Fig 3) that make it difficult to properly interpret the contribution of the cystic fibrosis phenotype versus

the innate or adaptive response to *Aspergillus*. This point might be addressed using bone marrow chimerism approaches, which would allow for the epithelial dysfunction phenotype but a normal immune phenotype. At the least, this would help to define if the epithelial injury proposed in relation to Figure 3 is responsible for these basal enhancements in the model being used.

2) This basal change has impact for several conclusions made, for example in L122 where the authors state that the Th9 response was sustained in the *cftr*^{-/-} mice. Instead, it may simply be that the mice exhibited a more robust response than the WT and failed to resolve as quickly.

3) The data in Figure 2h is concerning since it seems to counter the model being proposed, in which ILC2 activation supports the Th9 response. If this is so, the addition of IL-33 should surely have enhanced the IL-9 and transcription factor expression, which it does not seem to have done. Since IL-33 did not have any effects on the mast cell activation response shown in Figure 3, it becomes a concern that 1) their IL-33 was not degraded or inactive, and 2) the dose investigated was insufficient.

4) Assuming the model whereby IL-9 activates mast cells to produce IL-2, this alters the ILC2 and thereby drives Th9 is correct (as looks to be proposed), the key links between the ILC2 activation and Th9 responses remains unanswered.

5) The authors frame the role of IL-33 in the context of epithelial damage leading to release but this thinking is a little out of date. McKenzies group demonstrated that the epithelial cells of the lung that are IL-33 expressing at the type 2 pneumocytes and not the bronchial epithelial cells that the authors show disruption of in Figure 3.

6) The use of the *W-sh* mice in this study do provide issues. In particular, as the authors seem to allude to in their supplementary figure, these mice exhibit a stronger neutrophilic response that would influence the fungal load. Ganeshan et al. have shown that mast cells regulate neutrophil apoptosis in the lungs and that these mice have increased neutrophilia because of alternations in local survival.

7) The clinical data, while intriguing for the SNP, fails to convince for the IL-9 production in panel B. The variability would seem to suggest that more patients need to be done and that there are some high and low subgroups that might need to be considered. Also, some evidence that this SNP functionally alters transcription of the IL-9 gene would be useful, perhaps in the context of female hormones?

Minor issues:

P5, L165 typo. Should read "not".

P6, L184 typo. Should read "The IL-9..."

Point-by-point replay

Reviewer #1 (Remarks to the Author):

This is a topical and interesting area of study, as Th9 cells are a relatively a new subset, with many outstanding questions to be addressed. Here, Moretti et al. reported an interesting positive feedback loop amongst mast cells, ILC2 and Th9 cells to explain sustained IL-9 production in patients with cystic fibrosis. As presented, there are significant concerns with this paper, and some of the experiments are questionable and data may not their conclusions.

Major concerns:

1. The authors stated that the late IL-9 production is T cell-dependent (Line 98-99), but experiments demonstrating this claim are incomplete. At the very least, FACS plots showing lung CD4 T cells or Lin- ILC2 that produce IL-9 are important, simply using non-Th9 specific transcriptional factors such as PU.1 and IRF4, or IL-9 production in a pool of lung cells are insufficient to define Th9 cells. Furthermore, mRNA assessment in a pool of lung cells does not truly define specific functions of a subset of Th cells.

Response. The reviewer may be right. However, Figure 1 shows that IL-9 production late in infection is ablated in Rag1^{-/-} mice. This observation along with that of Supplementary Fig. 1 clearly shows that late IL-9 production is T cell-dependent. We have nevertheless incorporated in the new figure 1 FACS and immunofluorescence data showing the relative expression of IL-9 by ILC2 and CD4⁺T cells in vivo and ex-vivo during infection. Moreover, we have made it clear in the respective figure legend that the transcription factor Rora (and now also Gata3) has been evaluated on lineage negative cells and Th9 transcripts, Pu.1 and Irf4, and now Il9, on purified CD4⁺ T cells.

2. The manuscript would be more informative if the authors could provide absolute numbers of Lung ILC2 cells (Figure 1b), instead of only percentage, which can be misleading. The authors mentioned many times that CD25⁺ ILC2 expansion in Cfr^{-/-} mice, but CD25 is expressed by many other cell types also.

Response. We have provided the absolute number of lung ILC2 in Figure 1c. We are much aware that CD25 is expressed by many cell types. However, we are referring to the relative percentage of CD25⁺CD90.2⁺ double positive lung cells as clearly indicated in the x axis.

3. Upon Aspergillus fumigatus infection, the percentage of ILC2 is decreased dramatically (Fig 1b), while the transcriptional factor RoRa, which is believed to ILC2, is increased in WT C57BL/6 mice (Fig 1c). This paradox is not explained in the paper.

Response. The reviewer is correct. While the apparent increase in Rora mRNA expression is not statistically significant, we have added the Gata3 expression (new Figure 1d) that better correlates with the ILC2 dynamics.

4. In Figure 2C, the immune staining of the lung sections by CD90.2 and CD25 or ST2 is insufficient to define ILC2 cells. Activated T cells, even Tregs, are CD25⁺, and should be stained also. However, IL9R KO lung sections are completely negative for CD25 staining, especially in mice infected with Aspergillus fumigatus where T cell activation is expected.

Response. The reviewer is correct. Being IL-9R-deficient mice deficient of the IL-9Ra chain, they are also defective in CD25 expression. This minimizes the risk of assessing other CD25⁺ cells.

5. In Figure 2h, several key controls are missing. The IL-2 and IL-33 only groups should be included to rule out any direct effect of IL-2 and IL-33 on CD4 T cells.

Response. The reviewer is right. However, the key controls were and still are in Supplemental Figure 4. We have maintained these data in Supplemental Figure 4 for clarity.

6. In Figure 3e, in the ELISA assays performed for IL-9 production in cultures where exogenous IL-9 (100ng/ml) was added, how can you distinguish IL-9 produced from IL-9 added?

Response. We can't. However, considering the order of magnitude of the IL-9 added (100ng/ml) versus that measured (<100 pg/ml), it is unlikely that the amount measured refers to the amount left over upon the addition. As a matter of fact, this did not happen in cultures from control cells. Ultimately, we have adopted a protocol used by Demoulin et al (Oncogene. 2003; 22:1763-70) in which 500 Units/ml (corresponding to 100 ng) of IL-9 rapidly activated myeloid and lymphoid cells in vitro.

7. The authors state that MCs contribute to IL-2 production eventually leading to CD25⁺ ILC2 expansion (Line 169-170). But the experiments presented by no means show that IL-2 is from MC, not from activated T cells during infection. Does exogenous IL-2 rescue CD25⁺ ILC2 expansion in Figure 4a (or Line 175-176)?

Response. True. The new Figure 3 (panel f) now shows that MC, more than CD90.2 and CD4⁺T cells, produce IL-2 early in infection, particularly in Cftr^{-/-} mice. We agree with the reviewer that the relative contribution of each cell type to IL-2 production cannot be easily extrapolated. However, the lack of expansion of ILC2 in MC-deficient mice and its reversal by administration of either MC or IL-2 (see the new Figure 4, panel a), clearly point to an important, however not exclusive, role for IL-2 produced by MC. In addition, despite the visible expansion in Cftr^{-/-} mice, CD4⁺T cells producing IL-2 appear to be dispensable or redundant considering that CD25⁺ILC2 are, similar to C57BL/6 mice, expanded in Rag1^{-/-} mice.

8. Although the authors found a significant association between T allele of IL9 gene and IgE production in females, the patient number is very small, and caution should be used in making definitive conclusions.

Response. We have increased the patient number as much as we could and the new data are now included. We would also share with this reviewer the concern about the small sample size of our cohort of patients. On acknowledging this limitation, a new sentence in the discussion has been added.

Reviewer #2 (Remarks to the Author):

Moretti et al describe a mast cell/ILC2/Th9 circuit that regulates inflammation during fungal infection in wild type and CFTR-mutant mice. The authors show increased IL-9, ILC2, mast cell, and Th9 expansion during *A. fumigatus* infection, and that these responses are exaggerated in the Cftr^{-/-} mice. They show that IL-9/IL-9R is required for this response, and that IL-2 enhances Th9 cells co-cultured with ILCs and indicated by expression of IL-9, IRF4 and PU.1. They show that IL-9 stimulates IL-2 production from mast cells, and that the response is significantly increased in the absence of Cftr. Mast cells are required for this circuit because Kit-Wsh mice and imatinib-treated mice have lower cytokine production and diminished Th9 gene expression. Finally the authors

show sex specific effects of an IL9 polymorphism on IgE and IL-9 serum concentrations in CF patients.

This is a novel and interesting report. The authors have used a variety of approaches for defining this cellular circuit in models of fungal infection in wild type and CFTR-mutant mice. The inclusion of correlations in data from patients adds to the depth of the study. However, there are conclusions made that are not entirely supported, and some assumptions made that are not necessarily valid. The following are needed to complete this story.

1. It is presumed that when the authors examine PU.1 and IRF4 expression they are performing these assays in purified CD4 T cells, as both of these genes can be expressed in many other cell types. It would be important to show that these cells are actually making or capable of making IL-9. Minimally this could be done with the mRNA samples to examine IL9 expression. Optimally, seeing CD4⁺ T cells that are IL-9⁺ by flow cytometry would be more convincing.

Response. The reviewer is correct. We have incorporated in the new figure 1 FACS and immunofluorescence data showing the relative expression of IL-9 by ILC2 and CD4⁺T cells during infection. Moreover, the IL9 expression on purified CD4⁺ T cells has been added.

2. The authors show that mast cells when stimulated ex vivo can make IL-2 when stimulated with IL-9, but it is not clear this is occurring in vivo. The authors need to show flow cytometry of cells directly ex vivo (unstimulated or PMA/ionomycin) that are stained for IL-2 and then show how much of the IL-2 is coming from mast cells versus other cell types. They further need to show this is lacking in anti-IL-9 treated mice or IL9r^{-/-} mice, and increased in the Cftr-mutant mice. Evidence for this link in the circuit is lacking.

Response. Done. The new Figure 3 (panel f) incorporates now data on the comparative analysis of IL-2 production among MC, ILC2 and CD4⁺T cells by flow cytometry in C57BL/6 and Cftr^{-/-} mice. The new data show that MC, more than CD90.2 and CD4⁺T cells, produce IL-2 early in infection, particularly in Cftr^{-/-} mice. In addition, Figure 3f also shows that IL-2⁺MC failed to expand in IL9R^{-/-} mice, which is consistent with what stated in the text, i.e, that these mice have reduced IL-2. Ultimately, despite the visible expansion in Cftr^{-/-} mice, CD4⁺T cells producing IL-2 appear to be dispensable considering that CD25⁺ILC2 are, similar to C57BL/6 mice, expanded in Rag1^{-/-} mice.

3. In a similar point, the authors are not clearly showing the relative contribution of IL-9 from the various cell types. The authors seem to assume ILC2 cells are making IL-9 (Fig. 6), but this is never shown. Both CD4 T cells and mast cells are shown to make IL-9, but in separate assays where it's hard to compare production. At least at the level of mRNA (and optimally with flow cytometry) the authors need to provide a comparison of IL-9 production from the 3 cell types, probably at several time points, so a reader can appreciate what each is contributing to the environment.

Response. Figure 1 now shows that ILC2 (panel b) and CD4⁺ T cells (panel h) are making IL-9, early and late in the infection, respectively. For IL-9 production by MC, the new Figure 3, panel b, shows that MC produce IL-9 (also mentioned in the text). Although in separate figures, we hope that these new data clearly provide the contribution of each cell type to IL-9 production.

4. One point that the authors never address either experimentally or in the discussion is where the CFTR is having an effect. Is it in the cells defined within this circuit or in the lung cells that lead to different responses from the cell types. Obviously, this could be a whole new area, but I wonder if a reciprocal bone marrow

chimera approach to recapitulate data in Fig. 1 would at least determine if CFTR was affecting this circuit from hematopoietic cells versus airway or other structural cell.

Response. A totally new figure (Figure 5) and a new paragraph in the results is now devoted to the novel data obtained using bone marrow chimerism approaches.

5. It is not clear that the authors have purified ILCs when they examine Rora. As this gene is expressed in many cell types, this is critical. In general, the figure legends need to be clearer on what cells are being examined in assays for gene expression and cytokine production.

Response. Rora has been examined in lung lineage negative cells and the figure legend has been modified accordingly.

6. In Fig. 4a, the authors need to show, at least by graph, a comparison of the numbers of mast cells and ILCs between the B6 and Kit-Wsh mice. As shown, the data in that panel is hard to interpret.

Response. The graph has been added.

7. The title needs to be modified. The data from patient samples is strength, but it does not fully support the circuit, nor does it support a statement that this circuit impacts pathology. Similarly, the mouse model, while having the CF mutation, does not develop the same pathology as patients and the pathology being examined in this report is from acute fungal infection. The authors can decide how to handle this, but at least should include the phrase '....lung pathology in a mouse model of cystic fibrosis'.

Response. We agree with the reviewer. However, to avoid excessive penalization of the human aspect of our study, we have taken the liberty to modify the title as follows: "A Mast Cells/ILC2/Th9 pathway promotes lung inflammation in Cystic Fibrosis". Should this new title not entirely fulfilling the reviewer's suggestion, we are happy to modify the title as suggested.

8. The writing needs to be edited by a native English speaker; there were some parts that were hard to read and understand.

Response. Done.

Reviewer #3 (Remarks to the Author):

The paper by Moretti shows novel data on the role of IL-9 in influencing ILC2 and mast cell accumulation in response to Aspergillosis exposure - the data of which appears to have relevance in fungal sensitization in CF. The paper is well written and the data are novel and largely support the conclusions. There are a few issue with controls that need to be clarified.

1. In Figure 1, the fungal burden needs to be shown as the increased pathology in the CFTR^{-/-} mice could be due to greater antigen retention. Also total and fungal specific IgE responses, which is the hallmark of ABPA should be shown.

Response. The suggestion is correct. We have already shown (Am J Respir Crit Care Med 187:609-620) that the fungal growth was slightly increased in Cftr^{-/-} mice as compared to control mice. Similarly, we have shown high levels of total IgE in these mice. CFU and total serum IgE are now mentioned in the text.

2. It is unclear what the authors mean between infected vs ABPA in Figure 2b. This needs to be clarified. Again IgE levels should be shown here.

Response. We have specified in the text that mice were either acutely infected with Aspergillus conidia intranasally or subjected to ABPA by repeated sensitization with Aspergillus culture filtrate extracts, as described (Am J Respir Crit Care Med 187:609-620). The levels of IgE have been mentioned in the text.

3. Figure 3 should also include IgE. Can the sash mice make IgE but not just activate mast cells?

Response. IgE levels, albeit lower than control mice, were added to Figure 4b.

4. Did the authors assess Aspergillus specific IgE in the clinical cohort?

Response. Yes, we did. The results are now shown in the new Figure 5b.

Reviewer #4 (Remarks to the Author):

"A mast cell/ILC2/Th9 pathway promotes lung pathology in cystic fibrosis" by Moretti and colleagues examines the contribution of several effector mechanisms typically studied in allergic airway disease in the setting of cystic fibrosis. The key approach here is the use of the cftr^{-/-} mouse strain and studies on how these mice differ from controls in response to Aspergillus fumigatus infection. The authors reach the conclusion that there is a novel pathway through which mast cells coordinate with ILC2 cells to drive Th9 cell responses and pathogenesis. Importantly, the authors close the manuscript by showing associations of SNPs in the IL-9 locus with CF, which were rather intriguingly gender dependent.

In general, this study does provide aspects of data to support the fairly lengthy pathway that involves the three key cell types (mast cells, ILC2 and Th9) and the two key mediators (IL-2 and IL-9) but fails in the depth of inquiry to adequately determine if their conclusions are correct. The work also struggles in several places where there are points of the pathway that are not explained and/or the data seems to refute the mechanism to some degree. Consequently, the overall impression is that the work is an intriguing story but preliminary at this point.

Key specific areas of concern are as follows:

1) A concern that has impact on much of the data shown relates to the basal phenotype differences between the WT and the cftr^{-/-} strain. In several key figures, it is clear that there is a basal increase in many ILC2/Th2 cytokines and transcription factors (Fig 1C-F) and mast cells (Fig 3) that make it difficult to properly interpret the contribution of the cystic fibrosis phenotype versus the innate or adaptive response to Aspergillus. This point might be addressed using bone marrow chimerism approaches, which would allow for the epithelial dysfunction phenotype but a normal immune phenotype. At the least, this would help to define if the epithelial injury proposed in relation to Figure 3 is responsible for these basal enhancements in the model being used.

Response. A totally new figure (Figure 5) and a new paragraph in the results is now devoted to the novel data obtained using bone marrow chimerism approaches.

2) This basal change has impact for several conclusions made, for example in L122 where the authors state that the Th9 response was sustained in the *cftr*^{-/-} mice. Instead, it may simply be that the mice exhibited a more robust response than the WT and failed to resolve as quickly.

*Response. The reviewer is right. Due to the epithelium and myeloid dysfunction, *Cftr*^{-/-} mice have both a sustained Th9 response and a failure to resolve infection and inflammation.*

3) The data in Figure 2h is concerning since it seems to counter the model being proposed, in which ILC2 activation supports the Th9 response. If this is so, the addition of IL-33 should surely have enhanced the IL-9 and transcription factor expression, which it does not seem to have done. Since IL-33 did not have any effects on the mast cell activation response shown in Figure 3, it becomes a concern that 1) their IL-33 was not degraded or inactive, and 2) the dose investigated was insufficient.

*Response. Figure 2h clearly shows that the addition of IL-33 enhances the IL-9 and transcription factors expression in CD4⁺T cells co-cultivated with Lin⁻ cells particularly from *Cftr*^{-/-} mice. In addition, Supplementary Figure 4 now shows that the addition of IL-33, while not affecting Th9 transcription factors on CD4⁺T cells, significantly enhanced IL-9 production by Lin⁻ cells from WT and *Cftr*^{-/-} mice in the presence of *Aspergillus conidia*. Therefore, the ability of IL-33 to sustain an IL-9 response is not negated. Our data would suggest that at the increased levels observed in CF mice, IL-33 activates the ILC2/Th9 axis, with the contribution of mast cells, as opposed to WT mice in which the activity seems to be limited to ILC2 activation.*

4) Assuming the model whereby IL-9 activates mast cells to produce IL-2, this alters the ILC2 and thereby drives Th9 is correct (as looks to be proposed), the key links between the ILC2 activation and Th9 responses remains unanswered.

Response. Correct. We agree with the reviewer that mechanisms linking ILC2 activation to Th9 responses are far from being known. Our intention was merely to assess whether ILC2 promoted Th9 cell activation via IL-9R signaling. In this regard, the data of Figure 2h and Supplementary Fig. 4 clearly show that IL-2-responsive ILC2 are able to activate Th9 cells via IL-9R on responder CD4⁺ T cells. As such, although not pretending to explain all the mechanisms behind—not to mention the potential role of mast cells in Th9 cell polarization—these data are to our knowledge the first to show that IL-9R signalling on responder T cells is required for Th9 activation.

5) The authors frame the role of IL-33 in the context of epithelial damage leading to release but this thinking is a little out of date. McKenzies group demonstrated that the epithelial cells of the lung that are IL-33 expressing at the type 2 pneumocytes and not the bronchial epithelial cells that the authors show disruption of in Figure 3.

*Response. We were unable to replicate the observation of the McKenzies group. The figure below shows the results of the immunofluorescence staining of the lung from C57BL6 and *Cftr*^{-/-} mice, 3 days after the intranasal *Aspergillus* infection, clearly indicating that ST2⁺ epithelial cells are abundantly present.*

6) The use of the W-sh mice in this study do provide issues. In particular, as the authors seem to allude to in their supplementary figure, these mice exhibit a stronger neutrophilic response that would influence the fungal load. Ganeshan et al. have shown that mast cells regulate neutrophil apoptosis in the lungs and that these mice have increased neutrophilia because of alternations in local survival.

Response. We apologize for the typos. We have used MC-deficient C57BL/6-Kit^{W/W-v} mice that, despite being considered neutropenic (Am J Pathol. 2005:835-48) did mount a noticeable neutrophil response in infection. The mice are now properly cited in the text.

7) The clinical data, while intriguing for the SNP, fails to convince for the IL-9 production in panel B. The variability would seem to suggest that more patients need to be done and that there are some high and low subgroups that might need to be considered. Also, some evidence that this SNP functionally alters transcription of the IL-9 gene would be useful, perhaps in the context of female hormones?

Response. We have increased the patient number as much as we could and the new data are now included in the new Figure 5 (panel d). The database search (Haploreg v.4.1 and RegulomeDB) for transcriptional factor binding sites and regulatory regions failed to show the presence of hormonal-specific transcription factor binding sites at IL9 rs2069885. We may speculate that IL9 rs2069885, leading to a non-synonymous p.Thr117Met amino acid change in the IL9 protein, may affect the binding to its cognate receptor, IL9R. It is obvious that much additional studies are needed to shed light on this specific, however important, genetic aspect.

Minor issues:

P5, L165 typo. Should read "not".

Response. Done.

P6, L184 typo. Should read "The IL-9..."

Response. Done.

Reviewer #1 (Remarks to the Author):

The authors did invest substantial efforts in addressing my concerns, and the revised version is much better improved.

Reviewer #2 (Remarks to the Author):

The authors have satisfactorily addressed my previous concerns with new data and text. The addition of the bone marrow chimera experiment adds considerable depth to the studies that was lacking from the previous submission.

A few additional comments.

In discussing the bone marrow chimera experiments, the authors refer to myeloid and epithelial cells from the bone marrow. However, bone marrow should give rise to both lymphoid and myeloid cells. This is important because the lymphoid cells are clearly important for the model. This should be made clear in the description. The authors also note that the level of chimerism was examined, but they never state what that value was.

There are still a few places in the text where English is not optimal, but this could be fixed at the editorial level.

Reviewer #3 (Remarks to the Author):

The revised paper addressed most of my concerns. One minor issue is that the discussion needs to be modified to state that the human genetics of course will need to be replicated in an independent cohort - which is standard in the field.

Reviewer #4 (Remarks to the Author):

The authors have attempted to address the concerns but have failed in addressing some of the key points.

In particular, it is extremely clear that the *Cftr*^{-/-} mice have a basal elevation in almost every aspect of the proposed mechanism. The authors opt to ignore this and draw conclusions about the mechanisms of inflammation but it seems extremely likely that there is a predisposition to increased responses because of some altered tone in these mice.

In Figure 1 the authors use Gata3 as a marker for ILC2 but then in Figure 2 claim that it identifies the Th2 response. The authors really need to do a significantly more definitive assessment of the various cell types to draw any conclusions of this nature and simple mRNA increases in crude.

The point made in the response to reviewers regarding IL-9R deficient mice being defective in CD25 expression creates a concern for how the authors can then draw any conclusions regarding the CD25+ ILC2 cells in these mice.

The tryptase antibody used in Figure 3 is against human and the authors fail to demonstrate that this antibody is crossreactive (the company website also does not claim crossreactivity).

The bone marrow chimera experiment, while performed, does little to help remedy the concerns and the data seems to confuse the conclusions further.

Despite adding more patients, the sample size remains extremely underpowered for determining SNP differences between groups. Did the authors even perform power-calculations?

My concern regarding Figure 2h and the lack of effect of IL-33 remains unanswered. The authors response that there is a clear effect of IL-33, particularly on the Cftr^{-/-} Lin⁻ cells is not supported by the data, in which there is no significance indicated between the "None" and "IL-33" treatment groups except for an extremely small difference in IL-9 in the C57BL/6 (increasing from ~20pg/ml to ~30pg/ml).

My concern about the lack of mechanistic link between the ILC2 activation and Th9 responses is also unanswered in the response. Given that this is a central point of the overall study, it seems vital to elucidate in a meaningful manner and not simply ignore.

My concern about the source of IL-33 has been inadequately addressed and is a point that strongly impacts the overall hypothesis. As shown in the schematic, the authors are assuming that IL-33 is derived from damaged bronchial epithelial cells, a point that seems flawed when the IL-33 reporter mice findings are considered. Hardman et al. clearly showed type 2 cells were the dominant IL-33 expressing epithelial cells and that bronchial epithelial cells are not. It is unclear why the authors included a figure of ST2+ epithelial cells being increased in the Cftr^{-/-} mice because this furthers the confusion surrounding their overall schematic. What role precisely would ST2+ epithelial cells play in fungal responses in the Cftr^{-/-} mice?

The concept of mast cell production of IL-2 has been well-established, including in the setting of their activation by IL-33 (e.g see Morita et al. *Immunity*, 2015). There are technical issues with some of the approaches used to determine the conclusions. For example, mast cell deficient mice have been shown to have a deficit in ILC2 cells (see Russi et al. *J Immunol* 2015)

Point-by-point replay

Reviewers' comments:

Reviewer #1 (Remarks to the Author):

The authors did invest substantial efforts in addressing my concerns, and the revised version is much better improved.

Response. We thank the reviewer for the appreciation of our work.

Reviewer #2 (Remarks to the Author):

The authors have satisfactorily addressed my previous concerns with new data and text. The addition of the bone marrow chimera experiment adds considerable depth to the studies that was lacking from the previous submission.

A few additional comments.

In discussing the bone marrow chimera experiments, the authors refer to myeloid and epithelial cells from the bone marrow. However, bone marrow should give rise to both lymphoid and myeloid cells. This is important because the lymphoid cells are clearly important for the model. This should be made clear in the description. The authors also note that the level of chimerism was examined, but they never state what that value was.

Response. The text has been modified so to clarify both the points raised by the reviewer.

There are still a few places in the text where English is not optimal, but this could be fixed at the editorial level.

Response. We did our best to improve English.

Reviewer #3 (Remarks to the Author):

The revised paper addressed most of my concerns. One minor issue is that the discussion needs to be modified to state that the human genetics of course will need to be replicated in an independent cohort - which is standard in the field.

Response. The sentence has been corrected, accordingly (page 7).

Reviewer #4 (Remarks to the Author):

The authors have attempted to address the concerns but have failed in addressing some of the key points.

In particular, it is extremely clear that the *Cftr*^{-/-} mice have a basal elevation in almost every aspect of the proposed mechanism. The authors opt to ignore this and draw conclusions about the mechanisms of inflammation but it seems extremely likely that there is a predisposition to increased responses because of some altered tone in these mice.

Response. We have already argued that, due to the epithelium and myeloid dysfunction, Cftr^{-/-} mice have plausibly a basal sustained response, including Th9, and a failure to resolve infection and inflammation.

In Figure 1 the authors use *Gata3* as a marker for ILC2 but then in Figure 2 claim that it identifies the Th2 response. The authors really need to do a significantly more definitive assessment of the various cell types to draw any conclusions of this nature and simple mRNA increases in crude.

Response. We would like to make it clear that Gata3 was assessed on purified lineage negative cell, in Figure 1 (i.e. ILCs) and on purified CD4⁺T cells in Figure 2, consistent with the notion that GATA-3 regulates both ILC2 and Th2 cells (Immunity, 2012 Oct 19;37:589-91). Nevertheless, to avoid confusion and to further distinguish between ILC2 and Th2 cells, we have replaced Gata3 with Stat6 in the new figure 2 and added new data in Supplementary Figure 1 clearly showing the failure to upregulate Gata3 and Stat6 as well as the failure to phosphorylate STAT5 in infection in conditions of T cell-deficiency (i.e. Rag1^{-/-} mice). We believe these data provide the required distinction between ILC2 and Th2.

The point made in the response to reviewers regarding IL-9R deficient mice being defective in CD25 expression creates a concern for how the authors can then draw any conclusions regarding the CD25⁺ ILC2 cells in these mice.

Response. A new sentence on page 5 incorporates now the reviewer's suggestion.

The tryptase antibody used in Figure 3 is against human and the authors fail to demonstrate that this antibody is crossreactive (the company website also does not claim crossreactivity).

Response. The tryptase antibody used in Figure 3 has been successfully used to stain the mouse small intestine (see the Abcam website).

The bone marrow chimera experiment, while performed, does little to help remedy the concerns and the data seems to confuse the conclusions further.

Response. We share the opinion of Reviewer #2 that our bone marrow chimera experiment adds considerable depth to our work.

Despite adding more patients, the sample size remains extremely underpowered for determining SNP differences between groups. Did the authors even perform power-calculations?

Response. Of course, yes (see text page 7 and Supplementary Table 5).

My concern regarding Figure 2h and the lack of effect of IL-33 remains unanswered. The authors response that there is a clear effect of IL-33, particularly on the *Cftr*^{-/-} Lin⁻ cells is not supported by

the data, in which there is no significance indicated between the “None” and “IL-33” treatment groups except for an extremely small difference in IL-9 in the C57BL/6 (increasing from ~20pg/ml to ~30pg/ml).

Response. Figure 2h clearly shows that the addition of IL-33 enhances the IL-9 and transcription factors expression in CD4⁺T cells co-cultivated with Lin⁻ cells particularly from Cftr^{-/-} mice. In addition, Supplementary Figure 4 now shows that the addition of IL-33, while not affecting Th9 transcription factors on CD4⁺T cells, significantly enhanced IL-9 production by Lin⁻ cells from WT and Cftr^{-/-} mice in the presence of Aspergillus conidia. Therefore, the ability of IL-33 to sustain an IL-9 response is not negated. Our data would suggest that at the increased levels observed in CF mice, IL-33 activates the ILC2/Th9 axis, with the contribution of mast cells, as opposed to WT mice in which the activity seems to be limited to ILC2 activation.

My concern about the lack of mechanistic link between the ILC2 activation and Th9 responses is also unanswered in the response. Given that this is a central point of the overall study, it seems vital to elucidate in a meaningful manner and not simply ignore.

Response. By no means we have ignored this suggestion. We have clearly admitted that our intention was merely to assess whether ILC2 promoted Th9 cell activation via IL-9R signaling and not the full elucidation of the mechanistic links between ILC2 activation and Th9 response. In this regard, the data of Figure 2h and Supplementary Fig. 4 clearly show that IL-2-responsive ILC2 are able to activate Th9 cells via IL-9R on responder CD4⁺ T cells. As such, although not pretending to explain all the mechanisms behind—not to mention the potential role of mast cells in Th9 cell polarization—these data are to our knowledge the first to show that IL-9R signalling on responder T cells is required for Th9 activation.

My concern about the source of IL-33 has been inadequately addressed and is a point that strongly impacts the overall hypothesis. As shown in the schematic, the authors are assuming that IL-33 is derived from damaged bronchial epithelial cells, a point that seems flawed when the IL-33 reporter mice findings are considered. Hardman et al. clearly showed type 2 cells were the dominant IL-33 expressing epithelial cells and that bronchial epithelial cells are not. It is unclear why the authors included a figure of ST2⁺ epithelial cells being increased in the Cftr^{-/-} mice because this furthers the confusion surrounding their overall schematic. What role precisely would ST2⁺ epithelial cells play in fungal responses in the Cftr^{-/-} mice?

Response. We have clearly provided evidence in the previous point-by-point reply letter that ST2⁺ epithelial cells are abundantly present in the lung from C57BL6 and Cftr^{-/-} mice, 3 days after the intranasal Aspergillus infection. We have also provided a plausible explanation for the role this ST2⁺ epithelial cells may have in Cftr^{-/-} mice (see page 9 of the Discussion).

The concept of mast cell production of IL-2 has been well-established, including in the setting of their activation by IL-33 (e.g see Morita et al. Immunity, 2015). There are technical issues with some of the approaches used to determine the conclusions. For example, mast cell deficient mice have been shown to have a deficit in ILC2 cells (see Russi et al. JImmunol 2015)

Response. Our data are fully consistent with the Morita's (see Ref. 10) and Russi's findings.

Reviewer #3 (Remarks to the Author):

The revised paper addressed the majority of the issues. The authors should discuss the Hardman paper that shows that type II pneumocytes may be an important source of IL-33 in this model.

Point-by-point reply

Reviewer' comment:

Reviewer #3

The revised paper addressed the majority of the issues. The authors should discuss the Hardman paper that shows that type II pneumocytes may be an important source of IL-33 in this model.

Response: The paper of McKenzie's group has been now mentioned in the Discussion and the relevant reference included.